# Deployment Efficient Reward-Free Exploration with Linear Function Approximation

**Zihan Zhang**
zihanz@cse.ust.hk
HKUST

**Yuxin Chen**
yuxinc@wharton.upenn.edu
University of Pennsylvania

**Jason D. Lee**
jasondlee88@gmail.com
UC Berkeley

**Simon S. Du**
ssdu@cs.washington.edu
University of Washington

**Lin F. Yang**
linyang@ee.ucla.edu
UCLA

**Ruosong Wang**
ruosongwang@pku.edu.cn
Peking University

## Abstract

We study deployment-efficient reward-free exploration with linear function approximation, where the goal is to explore a linear Markov Decision Process (MDP) without revealing the reward function, while minimizing the number of distinct policies implemented during learning. By "deployment efficient", we mean algorithms that require few policies deployed during exploration – crucial in real-world applications where such deployments are costly or disruptive. We design a novel reinforcement learning algorithm that achieves near-optimal deployment efficiency for linear MDPs in the reward-free setting, using at most $H$ exploration policies during execution (where $H$ is the horizon length), while maintaining sample complexity polynomial in feature dimension and horizon length. Unlike previous approaches with similar deployment efficiency guarantees, our algorithm's sample complexity is independent of the reachability or explorability coefficients of the underlying MDP, which can be arbitrarily small and lead to unbounded sample complexity in certain cases – directly addressing an open problem from prior work. Our technical contributions include a data-dependent method for truncating state-action pairs in linear MDPs, efficient offline policy evaluation and optimization algorithms for these truncated MDPs, and a careful integration of these components to implement reward-free exploration with linear function approximation without sacrificing deployment efficiency.

## 1 Introduction

In real-world reinforcement learning applications, deploying new policies often incurs significant cost. For example, in robotics [Kober et al., 2013], deploying a new policy requires hardware-level operations, which can involve lengthy delays. In medical settings [Almirall et al., 2012, 2014, Lei et al., 2012], frequent policy changes are unrealistic, as each deployment typically requires a separate approval process involving domain experts. Similarly, in recommendation systems [Theocharous et al., 2015], deploying a new policy can take weeks due to mandatory internal testing to ensure safety and effectiveness. In all these scenarios, while switching policies frequently—especially based on instantaneous data, as standard RL algorithms require—is infeasible, it is often possible to run

39th Conference on Neural Information Processing Systems (NeurIPS 2025).

many experiments in parallel once a policy is deployed. This highlights the need for RL algorithms that learn effective policies while minimizing the number of policy deployments.

Empirically, the notion of *deployment efficiency* was first proposed by Matsushima et al. [2020], while a formal definition of deployment complexity was recently introduced by Huang et al. [2022]. Intuitively, deployment complexity measures the total number of policy deployments by an RL algorithm, under the constraint that the interval between policy switches—i.e., the number of trajectories collected before switching—is fixed in advance. Under this notion, a line of recent work has developed provably efficient RL algorithms [Huang et al., 2022, Qiao et al., 2022, Qiao and Wang, 2022] in various settings. In the tabular case where the state space is discrete and of small size, Qiao et al. [2022] designed an RL algorithm with $O(H)$ policy deployments, where $H$ is the horizon length. Huang et al. [2022], Qiao and Wang [2022] studied deployment complexity in the context of RL with linear function approximation (i.e., linear MDP [Yang and Wang, 2019, Jin et al., 2023]). Specifically, their algorithms achieve sample complexity polynomial in the feature dimension $d$ and horizon length $H$, with deployment complexity of $O(dH)$ or $O(H)$. Huang et al. [2022] further showed that any RL algorithm for linear MDPs must incur a deployment complexity of at least $\tilde{\Omega}(H)$.[1]

Although the aforementioned works provide important insights into the deployment complexity of reinforcement learning for linear MDPs, achieving the nearly optimal $O(H)$ deployment complexity remains challenging. Existing algorithms that attain this guarantee either operate in the tabular setting [Qiao et al., 2022]—which is unsuitable for large or continuous state spaces—or rely on strong assumptions such as the *reachability assumption* [Huang et al., 2022] or the *explorability assumption* [Qiao and Wang, 2022]. Roughly speaking, these assumptions require that all directions in the feature space can be explored by some policy. Such conditions are quite restrictive and significantly limit the applicability of these algorithms. In particular, they typically assume a lower bound on a *reachability coefficient* $v_{\min}$, and the sample complexity of existing algorithms with $O(H)$ deployment complexity depends polynomially on $\frac{1}{v_{\min}}$. In the tabular setting, this assumption is equivalent to requiring that every state can be reached with a non-negligible probability by some policy. However, in general linear MDPs, the reachability coefficient can be arbitrarily small, rendering the sample complexity effectively infinite for such algorithms.

To address this limitation, we investigate the following fundamental question:

*Is it possible to design RL algorithms for linear MDPs that achieve nearly optimal deployment complexity and polynomial sample complexity, without relying on additional assumptions such as reachability or explorability?*

This question was explicitly raised in prior work [Huang et al., 2022, Qiao and Wang, 2022] and was left as an open problem. Huang et al. [2022] conjectured that achieving $O(H)$ deployment complexity would necessarily require additional structural assumptions like reachability or explorability.

**Our Contribution.** In this paper, we resolve the above question by designing a new algorithm for linear MDPs with deployment complexity $H$. Our algorithm achieves polynomial sample complexity for *any* linear MDP and does not rely on additional assumptions such as reachability or explorability. Moreover, it operates in the *reward-free exploration* setting [Jin et al., 2020, Wang et al., 2020a, Chen et al., 2022, Wagenmaker et al., 2022, Zhang et al., 2021b, Li et al., 2024, 2023], where the reward function is not revealed during the exploration phase. This reward-free property further enhances the practicality of our approach in settings where reward signals are unavailable or costly to obtain. An informal statement of our main theoretical guarantee is summarized in the following theorem.

**Theorem 1** (Informal version of Theorem 4). *For reward-free exploration in linear MDPs, there is an algorithm (Algorithm 1) with deployment complexity $H$ and sample complexity polynomial in $d$, $H$, $1/\epsilon$, and $\log(1/\delta)$, such that with probability $1 - \delta$, for all linear reward functions, the algorithm returns a policy with suboptimality at most $\epsilon$. Here, $d$ is the feature dimension and $H$ is the horizon length.*

Combined with the existing hardness result from Huang et al. [2022], our new result in the above Theorem provides a complete answer to the deployment complexity of RL for linear MDPs. It shows that additional assumptions such as reachability or explorability, previously conjectured to be necessary, are in fact *not* required to achieve nearly optimal deployment complexity.

---

[1]Throughout this paper, we use $\tilde{O}$ and $\tilde{\Omega}$ to suppress logarithmic factors.

Table 1: Comparison with the most related works.

|  | Sample Complexity | Deployment Complexity |
|---|---|---|
| Huang et al. [2022] | $\text{poly}\left(d, H, \frac{1}{\epsilon}, \log(\frac{1}{\delta}), \frac{1}{v_{\min}}\right)$ | $H$ |
| Zhao et al. [2023] | $\tilde{O}\left(\frac{d^2 H^3}{\epsilon^2}\right)$ | $\tilde{O}(dH)$ |
| **This work** | $\tilde{O}\left(\frac{d^{15} H^{15}}{\epsilon^5}\right)$ | $H$ |

## 2 Related Work

There is a large body of literature on the sample complexity of RL. We refer readers to Agarwal et al. [2019], Chi et al. [2025] for more thorough reviews, and focus on the most relevant work here.

**Deployment Efficiency and Other Notions of Adaptivity.** The notion of *deployment efficiency* was first proposed in the empirical work [Matsushima et al., 2020], while its formal definition was first defined by Huang et al. [2022]. Under this notion, Huang et al. [2022], Qiao et al. [2022], Qiao and Wang [2022] designed provably efficient RL algorithms in various settings. As mentioned ealier, in order to achieve a nearly optimal deployment complexity, existing algorithms either work in the tabular setting, or rely on additional reachability assumption or explorability assumption which we strive to avoid in this work. Zhao et al. [2023] designed deployment efficient RL algorithms for function classes with bounded eluder dimension. However, even for linear functions, the deployment complexity of the algorithm by Zhao et al. [2023] is $\tilde{O}(dH)$, which is far from being optimal.

The notion of deployment efficiency is closely related to the low switching setting [Bai et al., 2019, Zhang et al., 2020c, Gao et al., 2021, Kong et al., 2021, Qiao et al., 2022, Wang et al., 2021]. We refer readers to prior work [Huang et al., 2022, Qiao et al., 2022] for a detailed comparison between these two different notions. Roughly speaking, in the low switching setting, the agent decides whether to update the policy or not after collecting each trajectory. On the other hand, the notion of deployment efficiency requires the interval between policy switching to be fixed, and therefore, deployment efficient RL algorithms are easier to implement in practice. The low switching setting was also studied for other sequential decision-making problems including bandits [Abbasi-Yadkori et al., 2011, Cesa-Bianchi et al., 2013, Simchi-Levi and Xu, 2019, Ruan et al., 2021].

**Reward-free Exploration.** The notion of reward-free exploration was first proposed by Jin et al. [2020]. In this setting, the agent first collects trajectories from an unknown environment without any pre-specified reward function. After that, a specific reward function is given, and the goal is to use samples collected during the exploration phase to output a near-optimal policy for the given reward function. The sample complexity of reward-free exploration was studied and improved in a line of work [Kaufmann et al., 2021, Ménard et al., 2021, Zhang et al., 2020b] A similar notion called task-agnostic exploration was consider by Zhang et al. [2020a], Li et al. [2024, 2023]. For linear MDPs, the first polynomial sample complexity for reward-free exploration was obtained by Wang et al. [2020a]. Later, the sample complexity was improved by Zanette et al. [2020], Wagenmaker et al. [2022]. Reward-free exploration was also considered in other RL settings including linear mixture MDPs [Chen et al., 2022, Zhang et al., 2021a] and RL with non-linear function approximation [Chen et al., 2022].

**Technical Comparison with Existing Algorithms.** Finally, we compare our new algorithm with existing algorithms with $O(H)$ deployment complexity [Qiao et al., 2022, Qiao and Wang, 2022] from a technical point of view. A more detailed overview of our new technical ingredients is given in Section 4. To achieve $O(H)$ deployment complexity in the tabular setting, Qiao et al. [2022] applied absorbing MDP to ignore those "hard to visit" states. In this work, similar ideas are used, though we work in the linear MDP setting which is much more complicated and requires a more careful treatment. In order to design an algorithm with $O(H)$ deployment complexity in linear MDPs under the explorability assumption, Qiao and Wang [2022] showed how to solve a variant of G-optimal experiment design in an offline manner. In this work, we also use offline RL to build exploration policies in linear MDPs. However, the lack of the explorability assumption raises substantial more technical challenges which necessitates more involved algorithms and analysis.

# 3 Preliminaries

In this section, we introduce the basic definitions of MDPs and the assumptions used in our analysis. We use $\Delta(X)$ to denote the set of probability distributions over a set $X$, and $[N]$ to denote the set $\{1, 2, \ldots, N\}$ for a positive integer $N$.

**Episodic MDPs.** A finite-horizon episodic Markov Decision Process (MDP) is defined by the tuple $(\mathcal{S}, \mathcal{A}, r, P, H, s_{\text{ini}})$, where $\mathcal{S} \times \mathcal{A}$ denotes the state-action space, $r : \mathcal{S} \times \mathcal{A} \times [H] \to [0, 1]$ is the reward function,[2] $P : \mathcal{S} \times \mathcal{A} \times [H] \to \Delta(\mathcal{S})$ is the transition kernel, $H$ is the episode horizon, and $s_{\text{ini}} \in \mathcal{S}$ is the initial state.[3]

A policy $\pi = \{\pi_h : \mathcal{S} \to \Delta(\mathcal{A})\}_{h=1}^H$ is a collection of mappings from the state space $\mathcal{S}$ to probability distributions over the action space $\mathcal{A}$, one for each time step $h \in [H]$. We say that $\pi$ is a *deterministic policy* if $\pi_h(s)$ assigns probability one to a single action for all $h$ and $s$.

In each episode, the learner starts from the initial state $s_1 = s_{\text{ini}}$ and proceeds as follows: at step $h = 1, \ldots, H$, the learner observes the current state $s_h$, selects an action $a_h$ according to $\pi_h(s_h)$, receives a reward $r_h = r_h(s_h, a_h)$, and transitions to the next state $s_{h+1}$ according to the transition kernel $P_h(\cdot \mid s_h, a_h)$. Fixing a policy $\pi$, we define the $Q$-function and the value function as follows: $\forall (s, a) \in \mathcal{S} \times \mathcal{A}$, $h \in [H]$,

$$Q_h^\pi(s, a) := \mathbb{E}_\pi \left[ \sum_{h'=h}^H r_{h'} \,\Big|\, (s_h, a_h) = (s, a) \right], \quad V_h^\pi(s) := \mathbb{E}_\pi \left[ \sum_{h'=h}^H r_{h'} \,\Big|\, s_h = s \right].$$

The optimal $Q$-function and value function are defined by:

$$Q_h^*(s, a) := \max_\pi Q_h^\pi(s, a), \quad V_h^*(s) := \max_\pi V_h^\pi(s).$$

By the Bellman optimality conditions, we have,

$$V_h^*(s) = \max_a Q_h^*(s, a), \quad Q_h^*(s, a) = r_h(s, a) + \mathbb{E}_{s' \sim P_h(\cdot|s,a)}[V_{h+1}^*(s')].$$

**Linear Function Approximation.** We assume that both the reward function and the transition kernel lie within a known low-dimensional subspace, a setting commonly referred to as a *linear MDP* [Yang and Wang, 2019, Jin et al., 2023].

**Assumption 2** (Linear MDP [Jin et al., 2023])**.** *Let $\{\phi_h(s, a)\}_{(s,a) \in \mathcal{S} \times \mathcal{A}, h \in [H]}$ be a collection of known feature vectors such that $\max_{s,a} \|\phi_h(s, a)\|_2 \leq 1$. For each $h \in [H]$, there exist vectors $\theta_h \in \mathbb{R}^d$ and $d$ measures $\mu_h = (\mu_h^1, \mu_h^2, \ldots, \mu_h^d)$ over the state space $\mathcal{S}$, representing the reward and transition kernels respectively, such that:*

$$r_h(s, a) = \langle \phi_h(s, a), \theta_h \rangle, \qquad \forall (s, a) \in \mathcal{S} \times \mathcal{A}, \tag{1a}$$

$$P_h(\cdot \mid s, a) = \langle \phi_h(s, a), \mu_h(\cdot) \rangle, \qquad \forall (s, a) \in \mathcal{S} \times \mathcal{A}, \tag{1b}$$

$$\|\theta_h\|_2 \leq \sqrt{d}. \tag{1c}$$

*Moreover, we assume $\left\| \int_{s \in \mathcal{S}} v(s) d\mu_h(s) \right\|_2 \leq \sqrt{d}$ for any mapping $v$ from $\mathcal{S}$ to $[-1, 1]$.*

Under Assumption 2, both the reward function and the transition kernel are linear in a shared set of $d$-dimensional features. This structure enables effective dimensionality reduction, especially when $d \ll SA$.

**Reward-free Exploration.** We now introduce the framework of reward-free exploration. This setting consists of two phases: the *exploration phase* (see Algorithm 1) and the *planning phase* (see Algorithm 5). In the exploration phase, the learner interacts with the environment—without access to any reward signal—to collect a dataset $\mathcal{D}$. In the planning phase, given any reward function $\{r_h\}_{h \in [H]}$ satisfying Assumption 2, the learner is required to output an $\epsilon$-optimal policy with probability at least $1 - \delta$, where $\epsilon$ is the accuracy parameter and $\delta$ is the failure probability.

**Deployment-efficient Reward-free Exploration.** We now present the definition of deployment complexity for reward-free exploration.

---

[2]We assume the reward is deterministic for simplicity.

[3]We may also assume the initial state $s_1$ is drawn from some fixed but unknown distribution $d_{\text{ini}}$, which can be modeled by setting the transition from $s_{\text{ini}}$ to follow $d_{\text{ini}}$.

**Definition 3** (Huang et al. [2022]). *An algorithm is said to have* deployment complexity $K$ *in linear MDPs if the following holds: given an arbitrary linear MDP satisfying Assumption 2, and for any accuracy parameter $\epsilon > 0$ and confidence level $\delta \in (0, 1)$, the algorithm performs at most $K$ policy deployments and collects $L$ trajectories per deployment, subject to the following constraints:*

(a) *With probability at least $1 - \delta$, for any reward kernel $\{\theta_h\}_{h \in [H]}$ satisfying Assumption 2, the learner returns an $\epsilon$-optimal policy $\pi$ under this reward kernel, i.e.,*

$$\mathbb{E}_\pi \left[ \sum_{h=1}^H \phi_h^\top(s_h, a_h)\theta_h \right] \geq \max_{\pi'} \mathbb{E}_{\pi'} \left[ \sum_{h=1}^H \phi_h^\top(s_h, a_h)\theta_h \right] - \epsilon,$$

*where the expectation $\mathbb{E}_\pi$ is taken over trajectories $\{s_h, a_h\}_{h=1}^H$ generated by executing policy $\pi$.*

(b) *The number of trajectories per deployment, $L$, is polynomial in the problem parameters, i.e., $L = \mathrm{poly}\left(d, H, \frac{1}{\epsilon}, \log \frac{1}{\delta}\right)$. Moreover, $L$ must be fixed* a priori *and cannot be adjusted adaptively between deployments.*

**Notations.** For positive semidefinite (PSD) matrices $A$ and $B$, we write $A \preceq B$ if $B - A$ is PSD, i.e., $B$ dominates $A$. We define the truncation operator $\mathtt{T}(A, B)$ as

$$\mathtt{T}(A, B) := \sup\{\zeta \leq 1 : \zeta A \preceq B\} \cdot A, \tag{2}$$

which represents the largest scaling of $A$ that is still dominated by $B$. For each $h \in [H]$ and $v \in \mathbb{R}^\mathcal{S}$, we define $\theta_h(v) := \mu_h^\top v$, where $\mu_h$ is the transition kernel. We also denote by $\mathbf{1}_s$ the $|\mathcal{S}|$-dimensional one-hot vector with a 1 in the $s$-th position.

## 4 Technical Overview

In this section, we give an overview of the technical challenges behind achieving Theorem 1, together our new ideas for tackling these challenges.

**The Layer-by-layer Approach.** Similar to existing algorithms with $O(H)$ deployment complexity [Huang et al., 2022, Qiao et al., 2022, Qiao and Wang, 2022], our new algorithm is based on a layer-by-layer approach. For each layer $1 \leq h \leq H$, based on an offline dataset obtained during previous iterations, our algorithm designs an exploration policy (a mixture of deterministic policies) for layer $h$, collect an offline dataset using the exploration policy, and then proceed to the next layer. Since we only use a single exploration policy for each layer, and there are $H$ layers, the deployment complexity would consequently be $H$. Following such an approach, datasets obtained for previous layers will be used for the purpose of policy optimization and policy evaluation for later layers, and therefore, the dataset should be able to cover all directions in the feature space. Therefore, we must carefully design the exploration strategy, so that for any direction that can be reached by some policy, our exploration strategy could also reach that direction up to an appropriate competitive ratio. By repeatedly sample trajectories following the exploration strategy, we would get a dataset that is sufficient for the purpose of policy optimization and policy evaluation for later layers.

**Dealing with Infrequent Directions.** The main technical issue associated with the above approach, is that there could be directions that cannot be reached frequently by any policy. In such a case, it is unrealistic to require such a direction to be reachable by the exploration policy. Existing algorithms with $O(H)$ deployment complexity [Huang et al., 2022, Qiao and Wang, 2022] avoids such an issue by assuming that any direction can be reached sufficiently frequently by some policy, in which case designing an exploration policy that can reach any direction in the feature space is feasible. However, since we do not assume explorability or reachability of the underlying MDP as in prior work [Huang et al., 2022, Qiao and Wang, 2022], we must handle those infrequent directions carefully.

If one simply chooses to ignore such infrequent directions, the error accumulated for handling such directions would in fact blow up exponentially, rendering the final sample complexity exponential in the feature dimension $d$ or the horizon length $H$. In fact, such an issue occurs even in the simpler tabular setting. In the tabular setting, an infrequent direction is equivalent to a state-action pair unreachable by any policy, and in order to handle such states, prior work [Qiao et al., 2022] applied absorbing MDP to ignore those "hard to visit" states. More specifically, once the algorithm detects

some state unreachable by any policy, that state would be directed to a dummy state in the absorbing MDP. Since we only direct states that are hard to visit to dummy states, the error accumulated during the whole process would be additive as we have more layers, which gives a polynomial sample complexity. Indeed, this is a high-level approach of the algorithm in Qiao et al. [2022].

On the other hand, for the linear MDP setting without the reachability assumption, handling infrequent directions is much more complicated. In the tabular setting, designing exploration policies is relatively simple since we can simply plan a policy for each individual state. On the other hand, for the linear MDP setting, we need to build the exploration policy in an iterative manner. Given directions that can be reached by the current exploration policy, we need to set the reward function appropriately to encourage exploring currently unreachable directions. More concretely, suppose the $\Lambda = \mathbb{E}[\phi\phi^\top]$ is the information matrix induced by the current exploration policy, for each state-action pair $(s, a)$ with feature $\phi(s, a)$, the reward function $r(s, a)$ would be set to $\phi(s, a)^\top \Lambda^{-1} \phi(s, a)$. We then plan a new policy for the current quadratic reward function, and test whether new policy can indeed reach some new direction, both by utilizing the offline dataset. We proceed to the next layer if the algorithm can no longer find any new reachable direction. The total number of directions found during the whole process can be shown to be small, using a standard potential function argument based on the determinant of the information matrix. To test whether the new policy can indeed reach some new direction, we need to estimate its information matrix $\Lambda = \mathbb{E}[\phi\phi^\top]$, again by using the offline dataset.

Note that by assuming reachability or explorability of the feature space, we no longer need to build the exploration policy iteratively since the whole feature space can be reached and therefore one can resort to approaches based on optimal experiment design. Indeed, this is the main idea behind previous work [Qiao and Wang, 2022]. However, such an approach critically relies on reachability or explorability of the feature space, which is one of the main technical challenges we aim to tackle.

**Handling Bias Induced by Infrequent Directions.** As mentioned, we heavily rely on the offline dataset obtained in previous layers for the purpose the offline policy optimization (planning for the quadratic reward function) and offline policy evaluations (for estimating the information matrix). Moreover, since we do not assume reachability of the feature space, there are always directions that cannot be reached by the exploration policy, and therefore, it is impossible for the offline dataset to cover the whole feature space. Imperfect coverage of the offline dataset will introduce additional error when conducting policy optimization and policy evaluation, due to the bias induced by infrequent directions. Although the error accumulated during offline policy optimization can be handle relatively easily, since a global argument based on comparing the groundtruth MDP and the MDP after ignoring infrequent directions would suffice, the error accumulated during offline policy evaluation is much more severe since the estimated information matrices would be used for deciding the next quadratic reward function. If not handled properly, the error will accumulate multiplicatively as we proceed to the next layer, rendering the final sample complexity exponential. Again, we note that by assuming reachability or explorability of the feature space as in prior work [Qiao and Wang, 2022], such an issue will not occur since the offline dataset would cover the whole feature space.

To handle such an issue, our new idea is to make sure the error of offline policy evaluation for estimating information matrices is always *multiplicative w.r.t. the information matrix to be evaluated*. More specifically, during the evaluation algorithm, if we encounter some state-action pair with feature $\phi = \phi(s, a)$, to ensure a multiplicative estimation error, we would add $\phi\phi^\top$ to the evaluation result $\Lambda$ only when $\phi^\top \Lambda^{-1} \phi$ is small. However, this will introduce another chicken-and-egg situation: without knowing the groundtruth information matrix $\Lambda$, it is impossible to test whether $\phi^\top \Lambda^{-1} \phi$ is small or not. To handle this, we use another iterative process to estimate the information matrix. Initially, the information matrix is set to be the identity matrix. In each iteration, in order to test whether $\phi^\top \Lambda^{-1} \phi$ is small or not, we use the information matrix $\Lambda$ obtained in the previous iteration, adding up $\phi\phi^\top$ for those $\phi$ that passed the test to form the new information matrix, and proceed to the next iteration. We stop the whole iteration process if the two information matrices obtained in two consecutive iterations are close enough in a multiplicative sense. By using another potential function argument based on the determinant of the information matrix, it can shown that the iterative process stops with small number of rounds. Such an idea is another major technical contribution of this paper.

**Handling Dependency Issues by Independent Copies.** As discussed ealier, our final algorithm involves two iterative processes, and since the results of different iterations all rely on the same offline dataset, these results are subtly coupled with each other. Fortunately, such dependency issues are relatively easy to handle, as we can simply make independent copies of the offline dataset by

following the exploration policy and repeatedly sampling trajectories with fresh randomness. We denote each independent copy as a sub-dataset, which will be explained in more details in Section 5.

Our final algorithm is a careful combination of all ideas mentioned above.

# 5  Algorithms

In this section, we describe our algorithms for achieving Theorem 1. The parameter settings are postpone to Appendix A due to space limitation.

**Datapoint and Sub-dataset.** The typical approach for handling linear MDPs is to treat $\{\phi_h(s, a), \tilde{s}\}$ as a datapoint, where for a state $s$, $\tilde{s}$ is the next state obtained by taking action $a$ at level $h$. In our algorithm, we further assign a weight $w$ to each datapoint to balance its importance in the whole dataset. As a result, one datapoint in our algorithm has form $\{\phi_h(s, a), \tilde{s}, w\}$. We remark that the weight $w$ is determined immediately once $\{\phi_h(s, a), \tilde{s}\}$ is collected.

In our algorithm, we conduct linear regression for multiple times, each time using a group of $N$ independent datapoints. Here, $N$ is a parameter to be decided. We denote these $N$ independent data-points as a sub-dataset, which has form $\{\phi_{h,i} = \phi_h(s_i, a_i), \tilde{s}_{h,i}, \lambda_{h,i}\}_{i \in [N]}$. To keep the statistical independence between different linear regression instances, we collect multiple independent copies of sub-datasets, so that the data used by different linear regression instances are independent.

**Exploration phase: Algorithm 1.** In the exploration phase, our algorithm collects samples in a layer-by-layer manner, and each layer uses a single deployment. In each layer, we assume that enough information about previous layers has been learned and focus on learning the current layer. For the current layer, `Policy-Design` is called to design the exploration policy based on existing samples, and `Policy-Execution` is called to execute the exploration policy and collect new samples.

In each call of `Policy-Design`, there are $m$ offline policy optimization sub-problems (see Line 6 of Algorithm 2) and $m$ offline policy evaluation sub-problems (see Line 11 of Algorithm 2). As mentioned, we collect multiple independent copies of datasets, and use a group of independent copies datasets to solve each sub-problem. More precisely, we collect $(2m^2 + 1) \cdot H$ independent copies for each dataset to solve the $2mH$ sub-problems, where each dataset consists of $N$ datapoints. Due to page limitation, the detail about how to collect samples is deferred to Algorithm 7 in the appendix.

**`Policy-Design` (Algorithm 2).** Given datasets in the first $h - 1$ layers, now we consider learning the $h$-th layer. The learner first designs reward function with form $r_h(s, a) \leftarrow \min\{\phi_h^\top(s, a)\Lambda^{-1}\phi_h(s, a), 1\}$, where $\Lambda$ is the current information matrix. We hope to update $\Lambda$ as

$$\Lambda_{\text{new}} \leftarrow \mathbb{E}_{\pi_{\text{old}}}\left[\phi_h \phi_h^\top\right] + \Lambda_{\text{old}},$$

where $\pi_{\text{old}}$ is a near-optimal policy w.r.t. the reward $r_{\text{old}} = \min\{\phi_h^\top \Lambda_{\text{old}}^{-1}\phi_h, 1\}$. By iteratively running this process, we will obtain some $\Lambda$ so that $\max_\pi \mathbb{E}_\pi\left[\min\{\phi_h^\top \Lambda^{-1}\phi_h, 1\}\right]$ is small. However, as discussed in Section 4, due to the infrequent directions, it is inappropriate to add $\mathbb{E}_{\pi_{\text{old}}}\left[\phi_h \phi_h^\top\right]$ to $\Lambda$ directly. Here, we need to truncate the infrequent directions in the distribution $\pi_{\text{old}}$, and evaluate the truncated matrix with the offline datasets. Below we explain how to address this by Algorithm 3.

**`Matrix-Eval` (Algorithm 3).** In Algorithm 3, the input is a policy $\pi$ and a group of datasets. The goal is to truncate the infrequent directions under $\pi$, and evaluate the information matrix after the truncation. To describe the high-level ideas, we assume $D$ is an distribution over $\mathbb{R}^d$ and the goal is to truncated the infrequent direction under $D$. For simplicity, we assume that $D$ is known, so that one can compute $\Lambda = \mathbb{E}_D[\phi\phi^\top]$ and those infrequent directions $\phi$ such that $\phi^\top \Lambda^{-1}\phi$ is large. The next step is to re-scale $\phi$, i.e., replace $\phi$ with $w(\phi) \cdot \phi$ such that $w^2(\phi)\phi^\top \Lambda^{-1}\phi$ is small. However, after truncation, the new information matrix would be $\Lambda_{\text{new}} = \mathbb{E}_{\phi \sim D}[w^2(\phi)\phi\phi^\top] \preceq \Lambda$, which means that a frequent direction under $\Lambda$ might turn to be an infrequent direction under $\Lambda_{\text{new}}$. A straightforward idea is to repeat this process until $\Lambda$ converges to some fixed point. Let $F(\Lambda) = \mathbb{E}_{\phi \sim D}\left[\text{T}(\phi\phi^\top, c_1\Lambda)\right]$ where $c_1$ is the threshold for truncation and $\text{T}$ is the operator defined in (2). By iteratively applying $F(\cdot)$ and noting that $F(\cdot)$ is non-increasing and the set of bounded PSD matrices is compact, the sequence $\{F^{(n)}(\Lambda)\}_{n \geq 1}$ will converge to some $\Lambda^*$ so that $F(\Lambda^*) = \Lambda^*$, in which case no more truncation is needed and hence, infrequent directions no longer exist. One might be worried that the zero matrix is also a fixed point of $F(\cdot)$ in which case the truncation is meaningless. Fortunately, by choose $c_1$ properly large, we can show that $\Pr_{\phi \sim D}[\phi^\top(\Lambda^*)^{-1}\phi \geq c_1] = O(\epsilon)$, where *epsilon*

is the desired accuracy. This means only a small portion of directions are truncated. When $D$ is unknown, we could draw samples from $D$ to estimate $\mathbb{E}_D[\mathrm{T}(\phi\phi^\top, \Lambda)]$ and run the same iterative process. Incorporating this idea with linear regression, we devise Algorithm 3 and 4 to evaluate the truncated information matrix efficiently.

In the planning phase, we employ standard backward planning for linear MDPs (e.g., Algorithm 5 `Planning` and Algorithm 6 `Planning-R`). See Appendix D for more details.

**Computational Efficiency.** We remark that the time complexity of our algorithm is polynomial in $d, H, 1/\epsilon$ and the number of actions $A$. In comparison, the algorithm in Qiao and Wang [2022] is computationally inefficient, and the algorithm in Huang et al. [2022] suffers time complexity depending on the realization parameter. We refer the readers to Appendix E for more details.

---

**Algorithm 1** `Exploration`

---

1: **Initialization:** $\mathcal{D}_h \leftarrow \emptyset$, $\check{\Lambda}_h \leftarrow \mathbf{I}$ for $h \in [H]$;
2: **for** $h = 1, 2, \ldots, H$ **do**
3:    $\left\{\{\pi^{i,h}\}_{i=1}^m, \check{\Lambda}_h\right\} \leftarrow$ `Policy-Design` $\left(h, \{\mathcal{D}_\tau^h(j)\}_{\tau\in[h-1], j\in[2m^2]}, \{\check{\Lambda}_\tau\}_{\tau\in[h-1]}\right)$;
4:    // *Roll out the policy and collect the datapoints. Each $\mathcal{D}_h^\tau(j)$ constructs a sub-dataset for the h-th layer;*
5:    $\{\mathcal{D}_h^\tau(j)\}_{j\in[2m^2+1], \tau\in[H]} \leftarrow$ `Policy-Execution` $\left(h, \{\pi^{i,h}\}_{i=1}^m, \check{\Lambda}_h\right)$;
6: **end for**
7: **return**: $\{\mathcal{D}_h^h(2m^2+1)\}_{h\in[H]}$ and $\{\check{\Lambda}_h\}_{h\in[H]}$

---

**Algorithm 2** `Policy-Design`

---

**Input:** horizon $h \in [H]$, block matrices $\{\check{\Lambda}_\tau\}_{\tau\in[h-1]}$, sub-datasets $\{\phi_{\tau,i}(j), \tilde{s}_{\tau,i}(j), \lambda_{\tau,i}(j)\}_{i\in[N]}$ for $\tau \in [h-1]$ and $j \in [2m^2]$;
**Initialization:** $\Lambda_h^0 = \zeta \mathbf{I}$;
**for** $\ell = 1, 2, \ldots, m$ **do**
  $r_h^\ell(s,a) \leftarrow \min\{\phi_h(s,a)^\top (\Lambda_h^{\ell-1})^{-1} \phi_h(s,a), 1\}$ for all $(s,a)$;
  $r_\tau^\ell(s,a) \leftarrow 0$ for $\tau \neq h$ and all $(s,a)$;
  $\{\pi^\ell, v_h^\ell\} \leftarrow$ `Planning-R`$(h, r^\ell := \{r_\tau^\ell\}_{\tau\in[H]}, \{\phi_{\tau,i}(m^2+\ell), \tilde{s}_{\tau,i}(m^2+\ell), \lambda_{\tau,i}(m^2+\ell)\}_{i\in[N],\tau\in[h-1]}, \{s_{1,i}(m^2+\ell)\}_{i=1}^N, \{\check{\Lambda}_\tau\}_{\tau\in[h-1]})$;
  // *Let $Y_{\tau,i}(a:b)$ denote $\{Y_{\tau,i}(j)\}_{j=a}^b$ for $a \leq b$ for $Y = \phi, \tilde{s}, \lambda$ and $s_1$;*
  $\check{\mathcal{D}} \leftarrow \{\phi_{\tau,i}((\ell-1)m-1 : \ell m), \tilde{s}_{\tau,i}((\ell-1)m-1 : \ell m), \lambda_{\tau,i}((\ell-1)m-1 : \ell m)\}_{i\in[N], \tau\in[h-1]}$;
  // *Feed independent sub-datasets to `Matrix-Eval`;*
  $\{\bar{\Lambda}_h^\ell, \check{\Lambda}_h^\ell\} \leftarrow$ `Matrix-Eval`$(h, \{\check{\Lambda}_\tau\}_{\tau\in[h-1]}, \pi^\ell, \check{\mathcal{D}})$;
  $\Lambda_h^\ell \leftarrow \Lambda_h^{\ell-1} + \bar{\Lambda}_h^\ell$;
**end for**
**return:** $\{\pi^{i,h}\}_{i=1}^m$ and $\check{\Lambda}_h \leftarrow \Lambda_h^m$.

---

**Algorithm 3** `Matrix-Eval`

---

1: **Input:** horizon $h \in [H]$, block matrices $\{\check{\Lambda}_\tau\}_{\tau=1}^{h-1}$, policy $\pi$, sub-datasets
   $\{\phi_{\tau,i}(j), \tilde{s}_{\tau,i}(j), \lambda_{\tau,i}(j)\}_{\tau\in[h-1], i\in[N], j\in[m]}$;
2: $\Lambda \leftarrow \mathbf{I}$;
3: **for** $j = 1, 2, \ldots, m$ **do**
4:    // *Estimate the truncated matrix with independent sub-datasets;*
5:    $\hat{F}_0 \leftarrow$ `Truncated-Matrix-Eval` $\left(h, \pi, \{\check{\Lambda}_\tau\}_{\tau=1}^{h-1}, \Lambda, \{\phi_{\tau,i}(j), \tilde{s}_{\tau,i}(j), \lambda_{\tau,i}(j)\}_{\tau\in[h-1], i\in[N]}\right)$;
6:    **if** $\hat{F}_0 + \frac{\zeta}{2x}\mathbf{I} \succeq \frac{1}{2}\Lambda$ **then**
7:      **break** and **return** $\left\{\hat{F}_0 + \frac{\zeta}{2x}\mathbf{I}, \Lambda\right\}$;
8:    **else**
9:      $\Lambda \leftarrow \hat{F}_0$;
10:   **end if**
11: **end for**

---

---

**Algorithm 4** `Truncated-Matrix-Eval`

---

1: **Input:** horizon $h$, policy $\pi$, block matrices $\{\check{\Lambda}_\tau\}_{\tau=1}^{h-1}$, truncation matrix $\Lambda$ , sub-datasets $\{\phi_{\tau,i}, \tilde{s}_{\tau,i}, \lambda_{\tau,i}\}_{\tau\in[h-1],i\in[N]}$;

2: $\hat{F}_h(s) \leftarrow \mathtt{T}(\phi_h(s,\pi_h(s))\phi_h^\top(s,\pi_h(s)), f_1\Lambda)$ for $s \in \{\tilde{s}_{h-1,i}\}_{i\in[N]}$;

3: **for** $\tau = h-1, h-2, ..., ..., 1$ **do**

4:  $X_\tau \leftarrow \sum_{i=1}^N \lambda_{\tau,i}^2 \phi_{\tau,i}\phi_{\tau,i}^\top + z\mathbf{I}$;

5:  **for** $s \in \{\tilde{s}_{\tau-1,i}\}_{i\in[N]}$ **do**

6:    $\phi \leftarrow \phi_\tau(s, \pi_\tau(s))$;

7:    **if** $\phi^\top \check{\Lambda}_\tau^{-1} \phi \geq 1$ **then**

8:     $\hat{F}_\tau(s) \leftarrow \mathbf{0}$;

9:    **else**

10:     $\hat{F}_\tau(s) \leftarrow \phi^\top X_\tau^{-1} \sum_{i=1}^N \lambda_{\tau,i}^2 \phi_{\tau,i}\hat{F}_{\tau+1}(\tilde{s}_{\tau,i}) + 2x\Lambda$;

11:    **end if**

12:  **end for**

13: **end for**

14: **return** : $\hat{F}_0 := \hat{F}_1(s_{\text{ini}})$;

---

## 6 Analysis

In this section, we present the formal version of the main theorem and sketch its proof.

**Theorem 4.** *By running Algorithm 1, the learner collects samples so that with probability $1 - \delta$, for any reward kernel $\{\theta_h\}_{h\in[H]}$ satisfying Assumption 2, the learner can return an $\epsilon$-optimal policy $\pi$ with Algorithm 5, i.e.,*

$$\mathbb{E}_\pi \left[ \sum_{h=1}^H \phi_h^\top(s_h, a_h)\theta_h \right] \geq \max_{\pi'} \mathbb{E}_{\pi'} \left[ \sum_{h=1}^H \phi_h^\top(s_h, a_h)\theta_h \right] - \epsilon.$$

*Moreover, Algorithm 1 uses $O(H)$ deployments and $\tilde{O}\left(\frac{d^{15}H^{15}}{\epsilon^5}\right)$ samples.*

Although we achieve reachability-independent sample complexity, the current dependencies on $d, H$ and $1/\epsilon$ are far from being optimal, especially compared to the bound in Qiao and Wang [2022]. The reason is that the technical difficulty changes significantly when allowing dependency on the reachability parameter. The core challenge in deployment-efficient linear MDPs arises from the fact that the linear regression problem becomes ill-conditioned when the reachability parameter $\lambda$ is very small. In the reachability-dependent methods (e.g., Qiao and Wang [2022]), one can pay $O(1/\lambda^*)$ episodes to collect samples $\{\phi_i\}_{i\geq 1}$ such that the information matrix $\sum \phi_i\phi_i^\top$ is well-conditioned. Meanwhile, in the reachability-independent methods, we need to identify the ill-conditioned directions and avoid these directions in linear regression. This step would be even harder given the constraint in deployments, which requires offline evaluation of the information matrix.

*Proof of Theorem 4.* We first analyze the deployment complexity and sample complexity.

**Deployment complexity.** For each $h = 1, 2, \ldots, H$, there is one deployment in Line 5. Therefore, the number of deployments is $H$.

**Sample complexity.** Algorithm 1 calls Algorithm 2 $H$ times, each requiring $(2m^2 + 1)N$ trajectories, resulting in a total sample complexity of $H \cdot H \cdot (2m^2 + 1)N = \tilde{O}\left(\frac{d^{15}H^{15}}{\epsilon^5}\right)$.

To finish the proof, we use the following lemma to prove the optimality of the learned policy. See full proof in Appendix C.9 ☐

**Lemma 5.** *With probability $1 - \delta$, for any reward kernel $\theta \in \{\theta_h\}_{h=1}^H$ satisfying Assumption 2, $\mathtt{Planning}\left(\theta, \{\phi_{h,i}, \tilde{s}_{h,i}, \lambda_{h,i}\}_{i=1}^N\}_{h\in[H]}, \{\check{\Lambda}_h\}_{h\in[H]}\right)$ (see Algorithm 5) returns an $\epsilon$-optimal policy, where $\{\phi_{h,i}, \tilde{s}_{h,i}, \lambda_{h,i}\}_{i=1}^N\}_{h\in[H]}$ and $\{\check{\Lambda}_h\}_{h\in[H]}$ is the output of Algorithm 1.*

To prove Lemma 5, a central lemma is introduced as follows, which states that the output sub-dataset of Algorithm 1 could efficiently cover all policies.

**Lemma 6.** *Recall that $\check{\Lambda}_\tau$ is the block matrix output by* `Policy-Design` *in Line 3 in the $\tau$-th iteration for $\tau \in [h-1]$. With probability $1 - \frac{\delta}{2} - \frac{\delta}{2H}$, for any sub-dataset of Algorithm 1 for the $h$-th layer $\{\phi_{h,i}, \tilde{s}_{h,i}, \lambda_{h,i}\}_{i \in [N]}$, we have*

*(i).* $\max_\pi \Pr_\pi \left[ \phi_h^\top \check{\Lambda}_h^{-1} \phi_h > 1, \phi_\tau^\top \check{\Lambda}_\tau^{-1} \phi_\tau \leq 1, \forall \tau \in [h-1] \right] \leq \frac{\epsilon}{8H^2}$ *for all $h \in [H]$;*

*(ii).* $\sum_{i=1}^N \lambda_{\tau,i}^2 \phi_{\tau,i} \phi_{\tau,i}^\top + z\mathbf{I} \succeq \frac{N}{8m} \check{\Lambda}_h$ *for all $h \in [H]$;*

*(iii).* $\lambda_{h,i}^2 \phi_{h,i}^\top \check{\Lambda}_h^{-1} \phi_{h,i} \leq f_1$ *for all $h \in [H]$ and $i \in [N]$.*

In proving Lemma 6, we use induction to construct a truncated MDP with information matrices $\{\check{\Lambda}_\tau\}_{\tau \geq 1}$. The three conditions in Lemma 6 serve the following purposes:

(i). To properly bound the truncation probability.

(ii). To ensure each $\check{\Lambda}_\tau$ is well-covered.

(iii). To rescale each sample for compatibility with the current information matrix $\check{\Lambda}_\tau$.

The proof of Lemma 6 is postponed to Appendix C.1 due to space limitation.

# 7 Conclusion

In this work, we design a new RL algorithm whose sample complexity is polynomial in the feature dimension and horizon length, while achieving nearly optimal deployment complexity for linear MDPs. Moreover, our algorithm works under the reward-free exploration setting, and does not require any additional assumptions on the underlying MDP. In our new algorithm and analysis, we propose new methods to truncate state-action pairs in a data-dependent manner, and design efficient offline algorithms for evaluating information matrices. Given our new results, an interesting future direction is to generalize our new techniques to other RL problems. For example, for function classes with bounded eluder dimension [Wang et al., 2020b, Kong et al., 2021, Zhao et al., 2023] , it would be interesting to design RL algorithm with nearly optimal $O(H)$ deployment complexity and polynomial sample complexity without relying on any additional assumptions.

# Acknowledgments

YC is supported in part by the Sloan Research Fellowship, the ONR grant N00014-22-1-2354, and the NSF grant CCF-2221009. JDL acknowledges support of Open Philanthropy, NSF IIS-2107304, NSF CCF-2212262, ONR Young Investigator Award, NSF CAREER Award 2144994, and NSF CCF-2019844. SSD acknowledges the support of NSF IIS-2110170, NSF DMS-2134106, NSF CCF-2212261, NSF IIS-2143493, NSF CCF-2019844, NSF IIS-2229881, and the Sloan Research Fellowship.

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

## A  Additional Parameter Settings and Notations

Assume $d, H \geq 40$, $\epsilon \leq \frac{1}{40}$. Set $x = \frac{1}{1000d^2H}$, $f_1 = \frac{320dH^2}{\epsilon}$, $\zeta = \frac{\epsilon^5}{10000d^5H^{15}}$, $\xi = \left(\frac{\epsilon}{10d^2H^2}\right)^{10}$, $z = \frac{100000\epsilon^2}{d^4H^5}$, $m = \frac{32000d^4H^3}{\epsilon}$, $N = \frac{10^9 d^7 H^7 \log\left(\frac{dH}{\epsilon\delta}\right)}{\epsilon^3}$. For a symmetric matrix $A$ and a PSD matrix $B$, we write $|A| \preceq B$ iff $B + A \succeq 0$ and $B - A \succeq 0$. We also present a table of notations as follows.

Table 2: Additional Notations.

| Notation | Comments |
|---|---|
| $P_h(\cdot|s,a)$ | the transition probability for the triple $(h,s,a)$ |
| $r_h(s,a)$ | the reward expectation for the triple $(h,s,a)$ |
| $\phi_h(s,a)$ | the $d$-dimensional feature vector for the triple $(h,s,a)$ |
| $\mu_h$ | the probability transition kernel be such that $P_h(\cdot|,s,a) = \mu\phi_h(s,a)$ |
| $\theta_h(v)$ | the $d$-dimensional payoff vector defined as $\mu_h^\top v$ |
| $\mathrm{T}(\cdot,\cdot)$ | the truncation function |
| $N$ | the number of datapoints in one dataset |
| $\{\phi_\tau, \tilde{s}_\tau, \lambda\}$ | one sample from the $\tau$-th layer |
| $\{\phi_{\tau,i}, \tilde{s}_{\tau,i}, \lambda_{\tau,i}\}_{i=1}^N$ | an independent dataset from the $\tau$-th layer |
| $\zeta$ | the regularization parameter |
| $\xi$ | the discretization parameter |
| $\mathcal{E}_1(\phi, v)$ | the concentration event for $\phi$ and value $v$ w.r.t. an independent dataset |
| $\mathcal{E}_2(\phi, f)$ | the concentration event for $\phi$ and matrix value $f$ w.r.t. an independent dataset |

## B  Technical Lemmas

**Lemma 7** (General Equivalence Theorem in Kiefer and Wolfowitz [1960])**.** *For any bounded subset $X \subset \mathbb{R}^d$, there exists a distribution $\mathcal{K}(X)$ supported on $X$, such that for any $\epsilon > 0$, it holds that*

$$\max_{x \in X} x^\top \left(\epsilon\mathbf{I} + \mathbb{E}_{y \sim \mathcal{K}(X)}[yy^\top]\right)^{-1} x \leq d. \tag{3}$$

*Furthermore, there exists a mapping $\pi^{\mathsf{G}}$, which maps a context $X$ to a distribution over $X$ such that*

$$\max_{x \in X} x^\top (\epsilon\mathbf{I} + \mathbb{E}_{y \sim \pi^{\mathsf{G}}(X)}[yy^\top])^{-1}x \leq 2d.$$

*When $\mathrm{supp}(X)$ has a finite size, $\pi^{\mathsf{G}}(X)$ could be implemented within $\mathrm{poly}(|\mathrm{supp}(X)|, \log(1/\epsilon))$ time.*

**Lemma 8.** *Assume $0 \leq \kappa \leq 0.1$. Let $\Lambda^0 = \zeta\mathbf{I}$. For each $i \geq 1$, let $D^i$ be a distribution over $\mathbb{R}^d$ satisfying that*

$$\mathbb{E}_{\phi \sim D^i}\left[\min\left\{\mathrm{Trace}\left(\phi\phi^\top(\Lambda^{i-1})^{-1}\right), 1\right\}\right] \geq \kappa \tag{4}$$

*and*

$$\Lambda^i \succeq \Lambda^{i-1} + \mathbb{E}_{\phi \sim D^i}[\phi\phi^\top].$$

*Then we have that*

$$\log(\det(\Lambda^n)) - \log(\det(\Lambda^0)) \geq \frac{n\kappa}{4}$$

*for any $n \geq 1$.*

*Proof.* Fix $i \geq 1$. Note that (4) is equivalent to

$$\mathbb{E}_{\phi \sim D^i}\left[\min\{\phi^\top(\Lambda^{i-1})^{-1}\phi, 1\}\right] \geq \kappa.$$

Let $W := \mathbb{E}_{\phi \sim D^i}\left[\mathtt{T}(\phi\phi^\top, \Lambda^{i-1})\right] \preceq \mathbb{E}_{\phi \sim D^i}\left[\phi\phi^\top\right]$. By definition, it holds that $W \preceq \Lambda^{i-1}$ and $W + \Lambda^{i-1} \preceq 2\Lambda^{i-1}$. We then have that

$$
\begin{aligned}
\log(\det(\Lambda^i)) - \log(\det(\Lambda^{i-1})) &\geq \log(\det(\Lambda^{i-1} + W)) - \log(\det(\Lambda^{i-1})) \\
&= \log\left(\det(\mathbf{I} + (\Lambda^{i-1})^{-1/2}W(\Lambda^{i-1})^{-1/2})\right) \\
&= \log\left(\det\left(\mathbf{I} + (\Lambda^{i-1})^{-1/2}\mathbb{E}_{\phi \sim D}\left[\mathtt{T}(\phi\phi^\top, \Lambda^{i-1})\right](\Lambda^{i-1})^{-1/2}\right)\right) \\
&\geq \frac{1}{4}\mathbb{E}_{\phi \sim D^i}\left[\mathrm{Trace}(\mathtt{T}(\phi\phi^\top, \Lambda^{i-1})(\Lambda^{i-1})^{-1})\right] \\
&\geq \frac{\kappa}{4}.
\end{aligned}
$$

The proof is completed by taking sum over $i$ from $1$ to $n$.

$\square$

### B.1 Concentration Inequalities

**Lemma 9.** *Let $X_1, X_2, ..., X_n$ be a group of zero-mean matrices such that $-\Lambda \preceq X_i \preceq \Lambda$ with probability $1$ for all $i \in [N]$. Let $w_1, w_2, ..., w_n$ be a group of reals. With probability $1 - \delta$,*

$$
\sum_{i=1}^n w_i X_i \succeq -2\sqrt{\sum_{i=1}^n w_i^2 \log(2d/\delta)}\Lambda - 2\max_i |w_i| \log(2d/\delta)\Lambda
$$

$$
\sum_{i=1}^n w_i X_i \preceq 2\sqrt{\sum_{i=1}^n w_i^2 \log(2d/\delta)}\Lambda + 2\max_i |w_i| \log(2d/\delta)\Lambda.
$$

*Proof.* Without loss of generality, we assume $\Lambda = \mathbf{I}$. For $0 \leq t \leq \frac{1}{\max_i |w_i|}$, define

$$
E_k = \mathbb{E}\left[\mathrm{Trace}\left(\exp\left(t\sum_{i=1}^k w_i X_i - 2t^2 \sum_{i=1}^k w_i^2 \mathbf{I}\right)\right)\right].
$$

Then we have that

$$
\begin{aligned}
&\mathbb{E}\left[E_k | X_{1:k-1}\right] \\
&\leq \mathbb{E}\left[\mathrm{Trace}\left(\exp\left(\log\left(\mathbb{E}[\exp(tw_k X_k)|X_{1:k-1}]\right) + t\sum_{i=1}^{k-1} w_i X_i - 2t^2 \sum_{i=1}^k w_i^2 \mathbf{I}\right)\right)\right] \\
&= \mathbb{E}\left[\mathrm{Trace}\left(\exp\left(\log(\mathbb{E}[\exp(tw_k X_k)|X_{1:k-1}]) - 2t^2 w_k^2 \mathbf{I} + t\sum_{i=1}^{k-1} w_i X_i - 2t^2 \sum_{i=1}^{k-1} w_i^2 \mathbf{I}\right)\right)\right] \\
&\leq \mathbb{E}\left[\mathrm{Trace}\left(t\sum_{i=1}^{k-1} w_i X_i - 2t^2 \sum_{i=1}^{k-1} w_i^2 \mathbf{I}\right)\right] \\
&= E_{k-1},
\end{aligned}
$$

where the first inequality is by Lieb's inequality (see Theorem 3.2, Tropp [2012]) and the second inequality is by Fact 10, 11 and 12. As a result, we learn that $\mathbb{E}[E_n] \leq \mathbb{E}[E_0] = d$, which means that with probability $1 - \delta/2$, the maximal eigenvalue of $\sum_{i=1}^k w_i X_i$ is at most $2\sqrt{\sum_{i=1}^n w_i^2 \log(2d/\delta)} + 2\max_i |w_i| \log(2d/\delta)$. Similar arguments work for the other side. The proof is completed. $\square$

**Fact 10.** *Assume $X$ is a stochastic symmetric matrix and $-\mathbf{I} \preceq X \preceq \mathbf{I}$ and $\mathbb{E}[X] = 0$. It then holds that*

$$
\mathbb{E}[\exp(tX)] \preceq \exp(2t^2)\mathbf{I}
$$

*for any $0 \leq t \leq 1$.*

*Proof.* By definition, we learn that

$$\exp(tX) = \sum_{k=0}^{\infty} \frac{(tX)^k}{k!}.$$

Taking expectation we learn that

$$\mathbb{E}[\exp(tX)] = \mathbf{I} + \sum_{k=2}^{\infty} \frac{t^k \mathbb{E}[X^k]}{k!} \preceq \mathbf{I} + \sum_{k=2}^{\infty} \frac{t^k}{k!} \mathbf{I} \preceq \exp(2t^2)\mathbf{I}.$$

$\square$

**Fact 11.** *Assume $X$ and $Y$ are two positively definite matrices such that $X \preceq Y$. It then holds that $\log(X) \preceq \log(Y)$.*

*Proof.* Note that for any $m \geq 0$, it holds that

$$\log(X) = \log(X + m\mathbf{I}) - \int_0^m (X + t\mathbf{I})^{-1} dt,$$

$$\log(Y) = \log(Y + m\mathbf{I}) - \int_0^m (Y + t\mathbf{I})^{-1} dt.$$

Because $X \preceq Y$, it holds that $-(X + t\mathbf{I})^t \preceq -(Y + t\mathbf{I})^{-1}$ for any $t \geq 0$. Then for any $m \geq 0$,

$$\log(X) \preceq \log(Y) + \log(Y + m\mathbf{I}) - \log(X + m\mathbf{I}).$$

Fix $\lambda > 0$ and choose $m \geq \frac{1}{\lambda} \|Y\|_\infty$. We have that $\log(Y + m\mathbf{I}) \preceq \log(m(1 + \lambda))\mathbf{I}$ and $\log(X + m\mathbf{I}) \succeq \log(m(1 - \lambda))\mathbf{I}$. As a result, for any $\lambda > 0$, we learn that

$$\log(X) \preceq \log(Y) + \log\left(\frac{1 + \lambda}{1 - \lambda}\right)\mathbf{I},$$

which implies $\log(X) \preceq \log(Y)$.

$\square$

**Fact 12.** *Let $X, Y$ be two symmetric matrices and $X \preceq 0$. It then holds that*

$$\text{Trace}\left(\exp(X + Y)\right) \leq \text{Trace}\left(\exp(Y)\right).$$

*Proof.* It suffices to verify that $\text{Trace}((X + Y)^k) \leq \text{Trace}(Y^k)$ for each $k \geq 2$, which is a direct result from Löwner–Heinz theorem.

$\square$

## C   Missing Lemmas and Proofs

### C.1   Proof of Lemma 6

We will prove by induction over the layers. Fix $h \in [H]$ and assume the three conditions in Lemma 6 holds for the first $h - 1$ layers. To facilitate the presentation of the proof, we first introduce the notion of truncated MDP.

**Truncated MDP.**   We define the truncated MDP $M_{h-1}$ by redirecting all state-action pairs $(s, a)$ to a dummy state at level $\tau$ if $\phi_\tau(s, a)^\top \check{\Lambda}_\tau^{-1} \phi_\tau(s, a) > 1$ for $\tau \in [h - 1]$. More precisely, a trajectory $\{(s_\tau, a_\tau)\}_{\tau=1}^H$ under the original MDP $M$ is mapped to $\{(s_1, a_1), \ldots, (s_k, a_k), \mathbf{z}, \ldots, \mathbf{z}\}$ under $M_{h-1}$. Here $k \leq h - 1$ is the smallest integer such that $\phi_k^\top(s_k, a_k)\check{\Lambda}_k^{-1} \phi_k(s_k, a_k) > 1$ and $\mathbf{z}$ is the dummy state. If $\phi_k^\top(s_k, a_k)\check{\Lambda}_k^{-1} \phi_k(s_k, a_k) \leq 1$ for all $k \in [h - 1]$, the trajectory is unchanged.

In the following, we re-define $\mathbb{E}[\cdot]$ and $\Pr[\cdot]$ to be the expectation and probability under $M_{h-1}$. We verify the three conditions as follows.

**Condition (i).** By Lemma 15, with probability $1 - \frac{\delta}{8H}$, $\max_\pi \mathbb{E}_\pi \left[ \min\{\phi_h^\top \check{\Lambda}_h^{-1} \phi_h, 1\} \right] \leq \frac{\epsilon}{8H^2}$, which implies that $\max_\pi \Pr_\pi \left[ \phi_h^\top \check{\Lambda}_h^{-1} \phi_h > 1 \right] \leq \frac{\epsilon}{8H^2}$. The proof is finished by noting the above inequality in the truncated MDP $M_{h-1}$ is equivalent to (i).

**Condition (ii).** By Lemma 19, with probability $1 - \frac{\delta}{16H}$, it holds that $\sum_{i=1}^N \lambda_{h,i}^2 \phi_{h,i} \phi_{h,i}^\top + z\mathbf{I} \succeq \frac{N}{8m} \check{\Lambda}_h$ for all sub-datasets $\{\phi_{h,i}, \tilde{s}_{h,i}, \lambda_{h,i}\}_{i=1}^N$.

**Condition (iii).** To verify the third condition, it suffices to note the definition $\lambda_{h,j} = \min \left\{ \sqrt{\frac{f_1}{\phi_{h,i}^\top \check{\Lambda}_h^{-1} \phi_{h,j}}}, 1 \right\}$ (See Algorithm 7).

The proof is finished.

## C.2  Statement and Proof of Lemma 13

**Lemma 13.** *Fix* $h \in [H]$. *Recall* $x = \frac{1}{100d^2 H} \geq 60\sqrt{\frac{md \log\left(\frac{dH}{\epsilon\delta}\right)}{N}}$. *Define* $F_h(s) := \hat{F}_h(s) = \mathtt{T}(\phi_h(s, \pi_h(s)) \phi_h^\top(s, \pi_h(s)), f_1 \Lambda)$. *For* $\tau = h-1, h-2, \ldots, 1$, *we define* $F_\tau(s) = \mathbb{E}_{s' \sim P_{\tau,s,\pi_\tau(s)}}[F_{\tau+1}(s') \cdot \mathbb{I}[\phi_\tau^\top(s, \pi_\tau(s)) \check{\Lambda}_\tau^{-1} \phi(s, \pi_\tau(s)) \leq 1]]$ *and* $F_0 = F_1(s_1) = F_1(s_{\text{ini}})$.

*Let* $\hat{F}_0$ *be the output of the Algorithm 4 with input* $\Lambda$ *and a group of independent sub-datasets* $\{\phi_{\tau,i}, \tilde{s}_{\tau,i}, \lambda_{\tau,i}\}_{\tau \in [h-1], i \in [N]}$. *we have that*

$$(1 - 3Hx)F_0 \preceq \hat{F}_0 \preceq (1 + 3Hx)F_0 + 4Hx\Lambda.$$

*Proof.* It is obvious that $F_\tau(s)$ is PSD for any proper $\tau$ and $s$. We prove by induction that

$$(1 - 3(h-\tau)x)F_\tau(s) \preceq \hat{F}_\tau(s) \preceq (1 + 3(h-\tau)x)F_\tau(s) + 4(h-\tau)x\Lambda \tag{5}$$

for any $1 \leq \tau \leq h$ and $s \in \{\tilde{s}_{\tau-1,i}\}_{i \geq 1}$.

For $\tau = h$, we have that $\hat{F}_\tau(s) = F_\tau(s)$ for any $s \in \mathcal{S}$. Fix $\ell \geq 2$ and assume that (5) holds for $\tau = \ell$.

For $s$ such that $\phi_{\ell-1}(s, \pi_{\ell-1}(s)) \check{\Lambda}_{\ell-1}^{-1} \phi(s, \pi_{\ell-1}(s)) > 1$, we have that $\hat{F}_{\ell-1}(s) = F_{\ell-1}(s) = \mathbf{0}$, where (5) holds trivially. Below we assume $\phi_{\ell-1}(s, \pi_{\ell-1}(s)) \check{\Lambda}_{\ell-1}^{-1} \phi(s, \pi_{\ell-1}(s)) \leq 1$. Recall that $X_\tau = \sum_{i=1}^N \lambda_{\ell-1,i}^2 \phi_{\ell-1,i} \phi_{\ell-1,i}^\top + z\mathbf{I}$. By definition, we have that for $s \in \{\tilde{s}_{\ell-2,i}\}_{i \geq 1}$

$$\hat{F}_{\ell-1}(s) = \phi_{\ell-1}(s, \pi_{\ell-1}(s))^\top X_\tau^{-1} \sum_{i=1}^N \lambda_{\ell-1,i}^2 \phi_{\ell-1,i} \hat{F}_\ell(\tilde{s}_{\ell-1,i}) + 2x\Lambda$$

$$= \mathbb{E}_{s' \sim P_{\ell-1,s,\pi_{\ell-1}(s)}} \left[ \hat{F}_\ell(s') \right] + \Delta_{\ell-1}^{(1)}(s) + 2x\Lambda$$

$$= \mathbb{E}_{s' \sim P_{\ell-1,s,\pi_{\ell-1}(s)}} [F_\ell(s)] + \Delta_{\ell-1}^{(1)}(s) + \Delta_{\ell-1}^{(2)}(s) + 2x\Lambda$$

$$= F_{\ell-1}(s) + \Delta_{\ell-1}^{(1)}(s) + \Delta_{\ell-1}^{(2)}(s) + 2x\Lambda, \tag{6}$$

where

$$\Delta_{\ell-1}^{(1)}(s) = \phi_{\ell-1}(s, \pi_{\ell-1}(s))^\top X_\tau^{-1} \sum_{i=1}^N \lambda_{\ell-1,i}^2 \phi_{\ell-1,i} \hat{F}_\ell(\tilde{s}_{\ell-1,i}) - \mathbb{E}_{s' \sim P_{\ell-1,s\pi_{\ell-1}(s)}} \left[ \hat{F}_\ell(s') \right]$$

$$= \phi_{\ell-1}(s, \pi_{\ell-1}(s))^\top X_\tau^{-1} \sum_{i=1}^N \lambda_{\ell-1,i}^2 \phi_{\ell-1,i} \hat{F}_\ell(\tilde{s}_{\ell-1,i}) - \phi_{\ell-1}(s, \pi_{\ell-1}(s))^\top \mu_{\ell-1}^\top \hat{F}_\ell(\cdot); \tag{7}$$

$$\Delta_{\ell-1}^{(2)}(s) = \mathbb{E}_{s' \sim P_{\ell-1,s,\pi_{\ell-1}(s)}} \left[ \hat{F}_\ell(s) - F_\ell(s) \right]. \tag{8}$$

By the induction assumption, we have that

$$0 \preceq (1 - 3(h-\ell)x)F_\ell(s) \preceq \hat{F}_\ell(s) \preceq (1 + 3(h-\tau)x)F_\ell(x) + 4(h-\tau)x\Lambda \preceq 2\Lambda.$$

By Lemma 14, with probability $1 - \frac{\delta}{16mH^2}$ it holds that

$$\Delta^{(1)}_{\ell-1}(s) \preceq 60\sqrt{\frac{md\log(\frac{dH}{\epsilon\delta})}{N}}\Lambda \preceq 2x\Lambda; \tag{9}$$

$$\Delta^{(1)}_{\ell-1}(s) \succeq -60\sqrt{\frac{md\log(\frac{dH}{\epsilon\delta})}{N}}\Lambda \succeq -2x\Lambda. \tag{10}$$

For the second term $\Delta^{(2)}_{\ell-1}(s)$, by the induction condition, we have that

$$\Delta^{(2)}_{\ell-1}(s) \preceq 3(h-\ell)x\mathbb{E}_{s'\sim P_{\ell-1,s,\pi_{\ell-1}(s)}}[F_\ell(s')] + 4(h-\ell)x\Lambda$$
$$= 3(h-\ell)xF_{\ell-1}(s) + 4(h-\ell)x\Lambda; \tag{11}$$
$$\Delta^{(2)}_{\ell-1}(s) \succeq -3(h-\ell)x\mathbb{E}_{s'\sim P_{\ell-1,s,\pi_{\ell-1}(s)}}[F_\ell(s')]$$
$$= -3(h-\ell)xF_{\ell-1}(s). \tag{12}$$

Putting all together and noting that $x \leq \frac{1}{100dH}$, we learn that

$$\hat{F}_{\ell-1}(s) - F_{\ell-1}(s) = \Delta^{(1)}_{\ell-1}(s) + \Delta^{(2)}_{\ell-1}(s) + 2x\Lambda$$
$$\preceq 2x\Lambda + (3(h-\ell)xF_{\ell-1}(s) + 4(h-\ell)x\Lambda)$$
$$\preceq 3(h-\ell+1)xF_{\ell-1}(s) + 4(h-\ell+1)x\Lambda \tag{13}$$
$$\hat{F}_{\ell-1}(s) - F_{\ell-1}(s) = \Delta^{(1)}_{\ell-1}(s) + \Delta^{(2)}_{\ell-1}(s) + 2x\Lambda$$
$$\succeq -x\Lambda - 3(h-\ell)xF_{\ell-1}(s) + 2x\Lambda$$
$$\succeq -3(h-\ell+1)xF_{\ell-1}(s); \tag{14}$$

The proof of (5) is finished.

Note that

$$\hat{F}_0 - F_0 = \hat{F}_1(s_{\text{ini}}) - F_1(s_{\text{ini}}).$$

Using the induction condition, for any $s \in \mathcal{S}$ it holds that

$$0 \preceq (1 - 3(H-1))F_1(s) \preceq \hat{F}_1(s) \preceq (1 + 3(H-1)x)F_1(s) + 4(H-1)x\Lambda \preceq 2\Lambda$$

As a result,

$$\hat{F}_1(s_{\text{ini}}) - F_1(s_{\text{ini}}) \preceq 3(h-1)xF_1(s_{\text{ini}}) + 4(h-1)x\Lambda$$
$$= 3(h-1)xF_0 + 4(h-1)x\Lambda;$$
$$\hat{F}_1(s_{\text{ini}}) - F_1(s_{\text{ini}})] \succeq -3(h-1)xF_1(s_{\text{ini}})$$
$$= -3(h-1)xF_0.$$

As a result, we obtain that

$$(1 - 3hx)F_0 \preceq \hat{F}_0 \preceq (1 + 3hx)F_0 + 4hx\Lambda.$$

The proof is finished.

$\square$

## C.3 Statement and Proof of Lemma 14

**Lemma 14.** *Fix $f : \mathcal{S} \to \mathbb{R}^{d^2}$ such that $0 \preceq f(s) \preceq \Lambda, \forall s \in \mathcal{S}$ for some PSD matrix $\Lambda$. Let $\{\phi_{\tau,i}, \tilde{s}_{\tau,i}, \lambda_{\tau,i}\}_{i=1}^N$ be a sub-dataset from the $\tau$-th layer. Assume $\{\phi_{\tau,i}, \tilde{s}_{\tau,i}, \lambda_{\tau,i}\}_{i=1}^N$ is independent of $f$. Let $X_\tau = \sum_{i=1}^N \lambda_{\tau,i}^2 \phi_{\tau,i}\phi_{\tau,i}^\top + z\mathbf{I}$. Then with probability $1 - \frac{\delta}{16mH^2}$*

$$\left| \phi^\top \mu_\tau^\top f - \phi^\top X_\tau^{-1} \sum_{i=1}^N \lambda_{\tau,i}^2 \phi_{\tau,i} f(\tilde{s}_{\tau,i}) \right| \preceq 60\sqrt{\frac{md\log\left(\frac{dH}{\epsilon\delta}\right)}{N}} \cdot \Lambda \tag{15}$$

*holds for any $\phi \in \mathbb{R}^2$ such that $\|\phi\|_2 \leq 1$ and $\phi^\top \check{\Lambda}_\tau^{-1}\phi \leq 1$.*

*Proof.* By the induction assumption (i) and (iii) about the sub-dataset $\{\phi_{\tau,i}, \tilde{s}_{\tau,i}, \lambda_{\tau,i}\}_{i=1}^{N}$ in Lemma 6, we have that $X_\tau \succeq \frac{N}{8m}\check{\Lambda}_\tau$ for $1 \leq \tau \leq h-1$ and $\max_i \phi_{\tau,i}^\top X_\tau^{-1}\phi_{\tau,i} \leq f_1$. By Lemma 17, with probability $1 - \frac{\delta}{16mH^2}$, we have that

$$\left| \phi^\top \mu_\tau^\top f - \phi^\top X_\tau^{-1} \sum_{i=1}^{N} \lambda_{\tau,i}^2 \phi_{\tau,i} f(\tilde{s}_{\tau,i}) \right|$$

$$\preceq \left( 16\sqrt{\phi^\top X_\tau^{-1}\phi\, d \log(\frac{dH}{\epsilon\delta})} + 8\sqrt{\max_i \phi_{\tau,i}^\top X_\tau^{-1}\phi_{\tau,i}\phi^\top X_\tau^{-1}\phi} \cdot d\log\left(\frac{dH}{\epsilon\delta}\right) + \zeta \right)\Lambda$$

$$\preceq 60\sqrt{\frac{md\log\left(\frac{dH}{\epsilon\delta}\right)}{N}} \cdot \Lambda.$$

$\square$

## C.4 Statement and Proof of Lemma 15

**Lemma 15.** *Recall the definition of $\check{\Lambda}_h = \Lambda_h^m$ in Algorithm 1. With probability $1 - \frac{\delta}{8H}$, it holds that*

$$\max_\pi \mathbb{E}_\pi \left[ \min\{\phi_h^\top \check{\Lambda}_h^{-1}\phi_h, 1\} \right] \leq \max\left\{ \frac{40d\log(3m/\zeta)}{m}, \frac{4}{3}B + \frac{2d}{f_1} \right\} \leq \frac{\epsilon}{8H^2}.$$

*Proof.* Recall the definition of $\{\Lambda_h^\ell\}_{\ell=0}^m$, $\{\bar{\Lambda}_h^\ell\}_{\ell=1}^m$ and $\{\check{\Lambda}_h^\ell\}_{\ell=1}^m$ in Algorithm 2. It then holds that $\Lambda_h^\ell = \Lambda_h^{\ell-1} + \bar{\Lambda}_h^\ell$ for $1 \leq \ell \leq m$. By the stop condition in Line 6, we have that $\bar{\Lambda}_h^\ell \succeq \check{\Lambda}_h^\ell$ for $1 \leq \ell \leq m$. Let $y^\ell = \max_\pi \mathbb{E}_\pi \left[ \min\{\phi_h^\top(\Lambda_h^\ell)^{-1}\phi_h, 1\} \right]$. Then $y^\ell$ is non-increasing in $\ell$ because $\Lambda_h^\ell$ is non-decreasing in $\ell$. Let $y = y^m = \max_\pi \mathbb{E}_\pi \left[ \min\{\phi_h^\top \check{\Lambda}_h^{-1}\phi_h, 1\} \right]$.

By Lemma 16 and Lemma 18, with probability $1 - \frac{\delta}{8mH} \cdot m = 1 - \frac{\delta}{8H}$,

$$\mathbb{E}_{\pi^\ell} \left[ \min\left\{ \mathrm{Trace}\left( \min\left\{ \frac{f_1}{\phi_h^\top(\check{\Lambda}_h^\ell)^{-1}\phi_h}, 1 \right\} \phi_h\phi_h^\top(\Lambda_h^{\ell-1})^{-1} \right), 1 \right\} \right]$$

$$\geq \mathbb{E}_{\pi^\ell} \left[ \min\{\mathrm{Trace}(\phi_h\phi_h^\top(\Lambda_h^{\ell-1})^{-1}), 1\} \right] - \Pr_{\pi^\ell}\left[ \phi_h^\top(\check{\Lambda}_h^\ell)^{-1}\phi_h > f_1 \right]$$

$$\geq \mathbb{E}_{\pi^\ell} \left[ \min\{\mathrm{Trace}(\phi_h\phi_h^\top(\Lambda_h^{\ell-1})^{-1}), 1\} \right] - \frac{d}{f_1(1-3Hx)}$$

$$\geq y^\ell - B - \frac{d}{f_1(1-3Hx)}$$

$$\geq y - B - \frac{d}{f_1(1-3Hx)}. \tag{16}$$

**Case i:** $y - B - \frac{d}{f_1(1-3Hx)} \geq \frac{y}{4}$. Recall that $\Lambda_h^\ell = \Lambda_h^{\ell-1} + \bar{\Lambda}_h^\ell$ for $1 \leq \ell \leq m$.
By Lemma 13 we have that

$$(1-3Hx)\mathbb{E}_{\pi^\ell}\left[ \min\left\{ \sqrt{\frac{f_1}{\phi_h^\top(\check{\Lambda}_h^\ell)^{-1}\phi_h}}, 1 \right\} \cdot \phi_h\phi_h^\top \right] \preceq \bar{\Lambda}_h^\ell.$$

On the other hand, by (16), we have that

$$(1-3Hx)\mathbb{E}_{\pi^\ell}\left[ \min\left\{ \mathrm{Trace}\left( \min\left\{ \frac{f_1}{\phi_h^\top(\check{\Lambda}_h^\ell)^{-1}\phi_h}, 1 \right\} \phi_h\phi_h^\top(\Lambda_h^{\ell-1})^{-1} \right), 1 \right\} \right] \geq \frac{(1-3Hx)y}{4} \geq \frac{y}{10}.$$

By Lemma 8 with the $D_\ell$ as the distribution of $\phi_h \cdot \sqrt{(1-3Hx)} \cdot \min\left\{ \sqrt{\frac{f_1}{\phi_h^\top(\check{\Lambda}_h^\ell)^{-1}\phi_h}}, 1 \right\}$ under $\pi^\ell$ and $\kappa = \frac{y}{10} \leq 0.1$, we have that

$$\log(\det(\Lambda_h^m)) - \log(\det(\Lambda_h^0)) \geq \frac{my}{40}. \tag{17}$$

Using Lemma 13, we have that $\bar{\Lambda}_h^\ell \preceq 3\mathbf{I}$ and thus $\log(\det(\Lambda_h^m)) \leq d\log(3m)$. On the other hand, we have that $\log(\det(\Lambda_h^0)) = d\log(\zeta)$, which means that $\frac{my}{40} \leq d\log(3m/\zeta)$. Therefore, we have that $y \leq \frac{40d\log(3m/\zeta)}{m} \leq \frac{\epsilon}{8H^2}$.

**Case ii:** $y - B - \frac{d}{f_1(1-3Hx)} < \frac{y}{4}$. In this case, we have that $y \leq \frac{4}{3}B + \frac{2d}{f_1} \leq \frac{\epsilon}{8H^2}$.

$\qquad\qquad\qquad\qquad\qquad\qquad\qquad\qquad\qquad\qquad\qquad\qquad\qquad\qquad\qquad\qquad\qquad\qquad\qquad$ $\square$

### C.5 Statement and Proof of Lemma 16

**Lemma 16.** *Let* $B = 2\sqrt{\frac{H^2\log(1/\delta)}{N}} + 2\frac{H\log(1/\delta)}{N} + 2H\left(32\sqrt{\frac{md\log\left(\frac{dH}{\epsilon\delta}\right)}{N}} + \frac{32md\sqrt{f_1}\log\left(\frac{dH}{\epsilon\delta}\right)}{N}\right)$.
*Let* $\{V_0^i, \pi^i\}$ *be the output of* Opt *with input reward as* $r^i$. *With probability* $1 - \frac{\delta}{8mH}$,

$$\max_\pi \mathbb{E}_\pi\left[r_h^i(s_h)\right] - \mathbb{E}_{\pi^i}\left[r_h^i(s_h)\right] \leq B.$$

*Proof.* Assume $w \in \mathbb{R}^\mathcal{S}$ satisfying $\|w\|_\infty \leq 1$. Let $\theta_\tau(w) = \mu_\tau^\top w$. By the induction condition $(i)$, we have that $X_\tau \succeq \frac{N}{8m}\check{\Lambda}_\tau$ for $\tau \in [h-1]$.

By Lemma 17 and the induction condition (iii) that $\lambda_{\tau,i}^2\phi_{\tau,i}^\top\check{\Lambda}_\tau^{-1}\phi_{\tau,i} \leq f_1$, with probability $1 - \frac{\delta}{16mH^2}$, we have that

$$\left|\phi^\top\theta_\tau(w) - \phi^\top X_\tau^{-1}\sum_{i=1}^N \lambda_{\tau,i}^2\phi_{\tau,i}\cdot\left(\phi_{\tau,i}^\top\theta_\tau(w) + \epsilon_i\right)\right|$$

$$\leq 8\sqrt{\phi^\top X_\tau^{-1}\phi\cdot d\log\left(\frac{dH}{\epsilon\delta}\right)} + 4\sqrt{\max_i\lambda_{\tau,i}^2\phi_{\tau,i}^\top X_\tau^{-1}\phi_{\tau,i}\cdot\phi^\top X_\tau^{-1}\phi\cdot d\log\left(\frac{dH}{\epsilon\delta}\right)} + \zeta$$

$$\leq 32\sqrt{\frac{md\log\left(\frac{dH}{\epsilon\delta}\right)}{N}} + \frac{32md\sqrt{f_1}\log\left(\frac{dH}{\epsilon\delta}\right)}{N} \tag{18}$$

for all $\phi$ such that $\|\phi\|_2 \leq 1$ and $\phi^\top\check{\Lambda}_\tau^{-1}\phi \leq 1$.

Let $\{v_\tau(s)\}$ and $\{v_\tau^*(s)\}$ denote respectively the value function under the policy $\pi^i$ and the optimal value function. Let $v_0 = v_1(s_{\text{ini}})$ and $v_0^* = \max_\pi \mathbb{E}_\pi\left[r_h^i(s_h)\right]$. Because $r_\tau^i(s,a) \in [0,1]$ for any proper $(s,a,\tau)$, we learn that $v_\tau(s), v_\tau^*(s), v_0, v_0^* \in [0,1]$. Recall the definition of $\{V_\tau(s)\}$ in Algorithm 6. We next prove by induction that $V_\tau(s) \geq v_\tau^*(s) \geq v_\tau(s)$ for any $s \in \mathcal{S}$ and $1 \leq \tau \leq h$. For $\tau = h$, the inequality is trivial. Assume $V_\tau(s) \geq v_\tau(s)$ for any $\ell \leq \tau \leq h$. By (18) with $w = V_\ell(\cdot)$

$$Q_{\ell-1}(s,a) \geq \mathbb{E}_{s'\sim P_{\ell-1,s,a}}[V_\ell(s')] \geq \mathbb{E}_{s'\sim P_{\ell-1,s,a}}[v_\ell^*(s')] \tag{19}$$

when $\phi_{\ell-1}^\top(s,a)\check{\Lambda}_{\ell-1}^{-1}\phi_{\ell-1}(s,a) \leq 1$. In the case $\phi_{\ell-1}^\top(s,a)\check{\Lambda}_{\ell-1}^{-1}\phi_{\ell-1}(s,a) > 1$, we have that

$$Q_{\ell-1}(s,a) = \mathbb{E}_{s'\sim P_{\ell-1,s,a}}[V_\ell(s')] = 0 \tag{20}$$

because $P_{\ell-1,s,a} = \mathbf{1}_\mathbf{z}$.

Therefore, we have that

$$V_{\ell-1}(s) = \text{Range}_{[0,1]}\left(\max_a Q_{\ell-1}(s,a)\right) \geq \text{Range}_{[0,1]}\left(\max_a \mathbb{E}_{s'\sim P_{\ell-1,s,a}}[v_\ell^*(s')]\right) = v_{\ell-1}^*(s).$$

By Bernstein's inequality, with probability $1 - \frac{\delta}{16mH}$, it holds that

$$V_0 = \frac{1}{N}\sum_{i=1}^N V_1(s_{1,i}) + 2\sqrt{\frac{H^2\log(1/\delta)}{N}} + 2\frac{H\log(16m/\delta)}{N} \geq V_1(s_{\text{ini}}) \geq v_1^*(s_{\text{ini}}) = v_0^*.$$

To bound the gap $\max_\pi \mathbb{E}_\pi \left[ r_h^i(s_h) \right] - \mathbb{E}_{\pi^i} \left[ r_h^i(s_h) \right]$, direct computation gives that

$$
\max_\pi \mathbb{E}_\pi \left[ r_h^i(s_h) \right] - \mathbb{E}_{\pi^i} \left[ r_h^i(s_h) \right]
$$

$$
= v_0^* - \mathbb{E}_{\pi^i} \left[ r_h^i(s_h) \right]
$$

$$
\leq V_0^i - \mathbb{E}_{\pi^i} \left[ r_h^{i-1}(s_h) \right]
$$

$$
= V_0^i - V_1(s_{\mathrm{ini}}) + \mathbb{E}_{\tau=1}^h \left[ V_\tau(s_\tau) - P_{\tau,s_\tau,a_\tau}^\top V_{\tau+1}(\cdot) \right]
$$

$$
\leq 2\sqrt{\frac{H^2 \log(1/\delta)}{N}} + 2\frac{H \log(1/\delta)}{N} + 2\sum_{\tau=1}^h \left( 32\sqrt{\frac{md \log\left(\frac{dH}{\epsilon\delta}\right)}{N}} + \frac{32md\sqrt{f_1} \log\left(\frac{dH}{\epsilon\delta}\right)}{N} \right) \quad (21)
$$

$$
= 2\sqrt{\frac{H^2 \log(1/\delta)}{N}} + 2\frac{H \log(1/\delta)}{N} + 2H \left( 32\sqrt{\frac{md \log\left(\frac{dH}{\epsilon\delta}\right)}{N}} + \frac{32md\sqrt{f_1} \log\left(\frac{dH}{\epsilon\delta}\right)}{N} \right)
$$

$$
= B,
$$

where (21) is by plugging $\phi_{\tau,s_\tau,a_\tau} = \phi$ and $w = V_{\tau+1}(\cdot)$ into (18):

$$
V_\tau(s_\tau) - P_{\tau,s_\tau,a_\tau}^\top V_{\tau+1}(\cdot) \leq 2 \left( 32\sqrt{\frac{md \log\left(\frac{dH}{\epsilon\delta}\right)}{N}} + \frac{32md\sqrt{f_1} \log\left(\frac{dH}{\epsilon\delta}\right)}{N} \right).
$$

$\square$

### C.6 Statement and Proof of Lemma 17

**Lemma 17.** *[Matrix concentration] Fix $v \in \mathbb{R}^{\mathcal{S}}$ such that $\|v\|_\infty \leq 1$ and $f : \mathcal{S} \to \mathbb{R}^{d^2}$ such that $0 \preceq f(s) \preceq \Lambda, \forall s \in \mathcal{S}$ for some $\Lambda$. Let $\{\phi_{\tau,i}, \tilde{s}_{\tau,i}, \lambda_{\tau,i}\}_{i=1}^N$ be a sub-dataset independent of $v$ and $f$ from the $\tau$-th layer. Let $X_\tau = \sum_{i=1}^N \lambda_{\tau,i}^2 \phi_{\tau,i} \phi_{\tau,i}^\top + z\mathbf{I}$. With probability $1 - \frac{\delta}{16mH^2}$, it holds that*

$$
\left| \phi^\top \theta(v) - \phi^\top X_\tau^{-1} \sum_{i=1}^N \phi_{\tau,i} v(\tilde{s}_{\tau,i}) \right|
$$

$$
\leq 8\sqrt{\phi^\top X_\tau^{-1} \phi (d \log(\frac{dH}{\epsilon\delta})} + 4\sqrt{\max_i \phi_{\tau,i}^\top X_\tau^{-1} \phi_{\tau,i} \phi^\top X_\tau^{-1} \phi} \cdot d \log(\frac{dH}{\epsilon\delta}) + \zeta.
$$

*and*

$$
\left| \phi^\top \mu^\top f - \phi^\top X_\tau^{-1} \sum_{i=1}^N \lambda_{\tau,i}^2 \phi_{\tau,i} f(\tilde{s}_{\tau,i}) \right|
$$

$$
\preceq \left( 16\sqrt{\phi^\top X_\tau^{-1} \phi d \log(\frac{dH}{\epsilon\delta})} + 8\sqrt{\max_i \phi_{\tau,i}^\top X_\tau^{-1} \phi_{\tau,i} \phi^\top X_\tau^{-1} \phi} d \log(\frac{dH}{\epsilon\delta}) + \zeta \right) \Lambda.
$$

*for any $\phi$ such that $\|\phi\|_2 \leq 1$.*

*Proof.* Let $\Phi(\xi)$ be an $\xi$-net of the $d$-dimensional unit ball w.r.t. $L_2$ norm. Recall that $\xi = \left( \frac{\epsilon}{10d^2 H^2} \right)^{10}$. Then $\log(\xi) \leq 20 \log(dH/\epsilon)$. Let

$$
\mathcal{E}_1(\phi, v)
$$

$$
:= \left\{ \left| \phi^\top \theta(v) - \phi^\top X_\tau^{-1} \sum_{i=1}^N \lambda_{\tau,i}^2 \phi_{\tau,i} v(\tilde{s}_{\tau,i}) \right| \leq 4\sqrt{\phi^\top X_\tau^{-1} \phi \log(1/\delta)} + 2\sqrt{\max_i \phi_{\tau,i}^\top X_\tau^{-1} \phi_{\tau,i} \phi^\top X_\tau^{-1} \phi} \cdot \log(1/\delta) \right\}.
$$

Then $\Pr[\mathcal{E}(\phi,v)] \leq 2\delta$ by Bernstein's inequality. Assume $\cup_{\phi \in \Phi(\xi)}\mathcal{E}_1(\phi,v)$ holds. Then for any $\phi \in \mathbb{R}^d$, letting $\psi$ be the nearest neighbor of $\phi$ in $\Phi(\xi)$, it holds that

$$\left| \phi^\top \theta(v) - \phi^\top X_\tau^{-1} \sum_{i=1}^N \phi_{\tau,i} v(\tilde{s}_{\tau,i}) \right|$$

$$\leq \left| \phi^\top \theta(v) - \psi^\top \theta(v) \right| + \left| \phi^\top X_\tau^{-1} \sum_{i=1}^N \phi_{\tau,i} v(\tilde{s}_{\tau,i}) - \psi^\top X_\tau^{-1} \sum_{i=1}^N \phi_{\tau,i} v(\tilde{s}_{\tau,i}) \right| + \left| \psi^\top \theta(v) - \psi^\top X_\tau^{-1} \sum_{i=1}^N \phi_{\tau,i} v(\tilde{s}_{\tau,i}) \right|$$

$$\leq \xi + \frac{N\xi}{z} + 4\sqrt{\psi^\top X_\tau^{-1} \psi \log(1/\delta)} + 2\sqrt{\max_i \phi_{\tau,i}^\top X_\tau^{-1} \phi_{\tau,i} \psi^\top X_\tau^{-1} \psi} \cdot \log(1/\delta)$$

$$\leq 4\sqrt{\phi^\top X_\tau^{-1} \phi \log(1/\delta)} + 2\sqrt{\max_i \phi_{\tau,i}^\top X_\tau^{-1} \phi_{\tau,i} \phi^\top X_\tau^{-1} \phi} \cdot \log(1/\delta) + \xi + \frac{N\xi}{z} + 6\log(1/\delta)\frac{2\xi}{z\sqrt{z}}$$

$$\leq 4\sqrt{\phi^\top X_\tau^{-1} \phi \log(1/\delta)} + 2\sqrt{\max_i \phi_{\tau,i}^\top X_\tau^{-1} \phi_{\tau,i} \phi^\top X_\tau^{-1} \phi} \cdot \log(1/\delta) + \zeta.$$

Noting that $|\Phi(\xi)| \leq (d/\xi)^d$, we have that $\Pr[\cup_{\phi \in \Phi(\xi)}]\mathcal{E}_1(\phi,v) \leq 2(d/\xi)^d \delta$. By replacing $\delta$ with $\frac{\delta}{16mH|\Phi(\xi)|}$, with probability $1 - 2\delta$, it holds that

$$\left| \phi^\top \theta(v) - \phi^\top X_\tau^{-1} \sum_{i=1}^N \phi_{\tau,i} v(\tilde{s}_{\tau,i}) \right|$$

$$\leq 4\sqrt{\phi^\top X_\tau^{-1} \phi \left( d + \log\left(\frac{d}{\xi\delta}\right) \right)} + 2\sqrt{\max_i \phi_{\tau,i}^\top X_\tau^{-1} \phi_{\tau,i} \phi^\top X_\tau^{-1} \phi} \cdot \left( d + \log\left(\frac{d}{\xi\delta}\right) \right) + \zeta.$$

for any $\phi$ such that $\|\phi\|_2 \leq 1$.

Define $\mathcal{E}_2(\phi,f)$ to be the event where

$$\left| \phi^\top \mu_\tau^\top f - \phi^\top X_\tau^{-1} \sum_{i=1}^N \lambda_{\tau,i}^2 \phi_{\tau,i} f(\tilde{s}_{\tau,i}) \right| \preceq \left( 4\sqrt{\phi^\top X_\tau^{-1} \phi \log(\frac{1}{\delta})} + 2\sqrt{\max_i \phi_{\tau,i}^\top X_\tau^{-1} \phi_{\tau,i} \phi^\top X_\tau^{-1} \phi} \log\left(\frac{1}{\delta}\right) \right) \Lambda$$

holds. We then show that $\Pr[\mathcal{E}_2(\phi,f)] \leq 2\delta$.

$$\phi^\top \mu_\tau^\top f - \phi^\top X_\tau^{-1} \sum_{i=1}^N \lambda_{\tau,i}^2 \phi_{\tau,i} f(\tilde{s}_{\tau,i}) = \phi^\top X_\tau^{-1} X_\tau \mu_\tau^\top f - \phi^\top X_\tau^{-1} \sum_{i=1}^N \lambda_{\tau,i}^2 \phi_{\tau,i} f(\tilde{s}_{\tau,i})$$

$$= \phi^\top X_\tau^{-1} \left( X_\tau \mu_\tau^\top f - \sum_{i=1}^N \lambda_{\tau,i}^2 \phi_{\tau,i} \left( \phi_{\tau,i} \mu_\tau^\top f + \epsilon_{\tau,i} \right) \right)$$

$$= -\sum_{i=1}^N \phi^\top X_\tau^{-1} \lambda_{\tau,i}^2 \phi_{\tau,i} \epsilon_{\tau,i} + \phi^\top X_\tau^{-1} z \mu_\tau^\top f, \quad (22)$$

where we define $\epsilon_{\tau,i} = \mathbb{E}_{s' \sim P_{\tau,s,a}}[f(s')] - f(\tilde{s}_{\tau,i})$ with $(s,a)$ being the state-action pair such that $\phi_\tau(s,a) = \phi_{\tau,i}$. Noting that $-\Lambda \preceq \epsilon_{\tau,i} \preceq \Lambda$ with probability 1, we have that

$$\sum_{i=1}^N \phi^\top X_\tau^{-1} \lambda_{\tau,i}^2 \phi_{\tau,i} \epsilon_{\tau,i}$$

$$\preceq 2\sqrt{\log(d/\delta) \cdot \sum_{i=1}^N \left( \lambda_{\tau,i}^2 \phi^\top X_\tau^{-1} \phi_{\tau,i} \right)^2 \Lambda + 2\max_i \left| \lambda_{\tau,i}^2 \phi^\top X_\tau^{-1} \phi_{\tau,i} \right| \log(d/\delta)\Lambda}$$

$$\preceq 2\sqrt{\log(d/\delta) \phi^\top X_\tau^{-1} \phi \Lambda} + 2\max_i \sqrt{\phi^\top X_\tau^{-1} \phi \cdot \lambda_{\tau,i}^2 \phi_{\tau,i}^\top X_\tau^{-1} \phi_{\tau,i} \Lambda} \quad (23)$$

holds with probability $1 - \delta$. In a similar way, with probability $1 - \delta$, we have

$$-\sum_{i=1}^{N} \phi^\top X_\tau^{-1} \lambda_{\tau,i}^2 \phi_{\tau,i} \epsilon_{\tau,i} \preceq 2\sqrt{\log(d/\delta) \phi^\top X_\tau^{-1} \phi} \Lambda + 2 \max_i \sqrt{\phi^\top X_\tau^{-1} \phi \cdot \lambda_{\tau,i}^2 \phi_{\tau,i}^\top X_\tau^{-1} \phi_{\tau,i}} \Lambda. \tag{24}$$

To bound the second term $z\phi^\top X_\tau^{-1} \mu_\tau^\top f$ in (22), we have

$$
\begin{aligned}
|z\phi^\top X_\tau^{-1} \mu_\tau^\top v| &\leq z\|\phi^\top X_\tau^{-1}\|_2 \|\mu_\tau^\top v\|_2 \\
&\leq \sqrt{z}\sqrt{z\phi^\top X_\tau^{-2}\phi} \cdot \sqrt{d} \\
&\leq \sqrt{zd \cdot \phi^\top X_\tau^{-1}\phi} \\
&\leq \sqrt{\phi^\top X_\tau^{-1}\phi}
\end{aligned}
\tag{25}
$$

for any $v \in \mathbb{R}^{\mathcal{S}}$ such that $\|v\|_\infty \leq 1$. As a result, we have $\|z\phi^\top X_\tau^{-1} \mu_\tau^\top\|_1 \leq \sqrt{\phi^\top X_\tau^{-1}\phi}$. Noting that $0 \preceq f(s) \preceq \Lambda$ for all $s \in \mathcal{S}$, we have that

$$-\sqrt{\phi^\top X_\tau^{-1}\phi} \Lambda \preceq z\phi^\top X_\tau^{-1} \mu_\tau^\top f \preceq \sqrt{\phi^\top X_\tau^{-1}\phi} \Lambda. \tag{26}$$

By (22), (23), (24) and (26), we have that

$$
\left| \phi^\top \mu_\tau^\top f - \phi^\top X_\tau^{-1} \sum_{i=1}^{N} \lambda_{\tau,i}^2 \phi_{\tau,i} f(\tilde{s}_{\tau,i}) \right|
$$
$$
\preceq 4\sqrt{\log(d/\delta) \phi^\top X_\tau^{-1} \phi} \Lambda + 2 \max_i \sqrt{\phi^\top X_\tau^{-1} \phi \cdot \lambda_{\tau,i}^2 \phi_{\tau,i}^\top X_\tau^{-1} \phi_{\tau,i}} \Lambda \tag{27}
$$

The proof is finished. Assume $\cup_{\phi \in \Phi(\xi)} \mathcal{E}_2(\phi, f)$ holds. Fix $\phi$ and let $\psi$ be the nearest neighbor of $\phi$ in $\Phi(\xi)$. We then have that

$$
\phi^\top \mu_\tau^\top f - \phi^\top X_\tau^{-1} \sum_{i=1}^{N} \phi_{\tau,i} f(\tilde{s}_{\tau,i})
$$
$$
= \left( \phi^\top \mu_\tau^\top f - \psi^\top \mu_\tau^\top f \right) + \left( \phi^\top X_\tau^{-1} \sum_{i=1}^{N} \phi_{\tau,i} f(\tilde{s}_{\tau,i}) - \psi^\top X_\tau^{-1} \sum_{i=1}^{N} \phi_{\tau,i} f(\tilde{s}_{\tau,i}) \right)
$$
$$
+ \left( \psi^\top \theta(v) - \psi^\top X_\tau^{-1} \sum_{i=1}^{N} \phi_{\tau,i} f(\tilde{s}_{\tau,i}) \right). \tag{28}
$$

We then bound the three terms in (28) separately. For the first term, we have that $|(\phi - \psi)^\top \mu_\tau^\top v| \leq \xi\sqrt{d}$ for any $v \in \mathbb{R}^{\mathcal{S}}$ such that $\|v\|_\infty \leq 1$. As a result, we have that $\|\mu_\tau(\phi - \psi)\|_1 \leq \xi\sqrt{d}$, which implies that

$$-\xi\sqrt{d}\Lambda \preceq \phi^\top \mu_\tau^\top f - \psi^\top \mu_\tau^\top f \preceq \xi\sqrt{d}\Lambda. \tag{29}$$

For the second term, we have that

$$
\left| \phi^\top X_\tau^{-1} \sum_{i=1}^{N} \phi_{\tau,i} v(\tilde{s}_{\tau,i}) - \psi^\top X_\tau^{-1} \sum_{i=1}^{N} \phi_{\tau,i} v(\tilde{s}_{\tau,i}) \right| \leq \frac{N\xi}{z}
$$

for any $v \in \mathbb{R}^{\mathcal{S}}$ such that $\|v\|_\infty \leq 1$. Using similar arguments, we learn that

$$
\left\| \phi^\top X_\tau^{-1} \sum_{i=1}^{N} \phi_{\tau,i} - \psi^\top X_\tau^{-1} \sum_{i=1}^{N} \phi_{\tau,i} \right\|_1 \leq \frac{\sqrt{d}N\xi}{z}
$$

and

$$-\frac{\sqrt{d}N\xi}{z}\Lambda \preceq \phi^\top X_\tau^{-1}\sum_{i=1}^N \phi_{\tau,i}f(\tilde{s}_{\tau,i}) - \psi^\top X_\tau^{-1}\sum_{i=1}^N \phi_{\tau,i}f(\tilde{s}_{\tau,i}) \preceq \frac{\sqrt{d}N\xi}{z}\Lambda. \tag{30}$$

By $\cup_{\phi \in \Phi(\xi)}\mathcal{E}_2(\phi, f)$, we could bound the third term as

$$\left|\psi^\top\theta(v) - \psi^\top X_\tau^{-1}\sum_{i=1}^N \phi_{\tau,i}f(\tilde{s}_{\tau,i})\right| \preceq 4\sqrt{\log(d/\delta)\psi^\top X_\tau^{-1}\psi}\Lambda + 2\max_i \sqrt{\psi^\top X_\tau^{-1}\psi\lambda_{\tau,i}^2\phi_{\tau,i}^\top X_\tau^{-1}\phi_{\tau,i}}\Lambda. \tag{31}$$

Putting (29), (30) and (31) together, we learn that

$$\left|\phi^\top\mu_\tau^\top f - \phi^\top X_\tau^{-1}\sum_{i=1}^N \phi_{\tau,i}f(\tilde{s}_{\tau,i})\right|$$

$$\preceq \left(\xi\sqrt{d} + \frac{\sqrt{d}N\xi}{z} + 4\sqrt{\log(d/\delta)\psi^\top X_\tau^{-1}\psi} + 2\max_i \sqrt{\psi^\top X_\tau^{-1}\psi\lambda_{\tau,i}^2\phi_{\tau,i}^\top X_\tau^{-1}\phi_{\tau,i}}\right)\Lambda$$

$$\leq \left(\xi\sqrt{d} + \frac{\sqrt{d}N\xi}{z} + \frac{12\log(d/\delta)\xi}{z\sqrt{z}} + 4\sqrt{\log(d/\delta)\phi^\top X_\tau^{-1}\phi} + 2\max_i \sqrt{\phi^\top X_\tau^{-1}\phi\lambda_{\tau,i}^2\phi_{\tau,i}^\top X_\tau^{-1}\phi_{\tau,i}}\right)\Lambda$$

$$\leq \left(4\sqrt{\log(d/\delta)\phi^\top X_\tau^{-1}\phi} + 2\max_i \sqrt{\phi^\top X_\tau^{-1}\phi\lambda_{\tau,i}^2\phi_{\tau,i}^\top X_\tau^{-1}\phi_{\tau,i}} + \zeta\right)\Lambda. \tag{32}$$

The proof is finished by replacing $\delta$ with $\frac{\delta}{16mH|\Phi(\xi)|}$.

$\square$

### C.7 Statement and Proof of Lemma 18

**Lemma 18.** *By running Algorithm 3, we have the following claims: (1) The iteration in line 3 ends in $10d\log\left(\frac{2x}{v} + 1\right)$ rounds; (2) Let $\Lambda_{\mathrm{end}}$ be the final value of $\Lambda$. Then it holds that*

$$\Pr_\pi\left[\phi_h^\top(\Lambda_{\mathrm{end}})^{-1}\phi_h > f_1\right] \leq \frac{d}{f_1(1 - 3Hx)}.$$

*Proof.* Fix $\pi$. Let $\hat{F}_0$ be the output of Algorithm 4 with input $(h, \{\check{\Lambda}_\tau\}_{\tau=1}^{h-1}, \Lambda, \mathcal{D})$ where $\mathcal{D}$ is a group of valid sub-datasets. Since $h$ and $\{\check{\Lambda}_\tau\}_{\tau=1}^{h-1}$ are fixed in the context, we write $\hat{F}_0 = \hat{F}_0(\Lambda)$ as a (stochastic) function of $\Lambda$. We also define the expected truncated matrix as

$$F_0(\Lambda) := \mathbb{E}_\pi\left[\mathrm{T}(\phi_h\phi_h^\top, f_1\Lambda) \cdot \mathbb{I}[\phi_\tau(s_\tau, \pi_\tau, s_\tau)^\top\check{\Lambda}_\tau^{-1}\phi_\tau(s_\tau, \pi_\tau, s_\tau) < 1, \forall 1 \leq \tau \leq h]\right].$$

**Number of iterations.** Let $\Lambda_i$ be the value of $\Lambda$ after the $i$-th iteration. Suppose there are $T$ iterations. For $1 \leq i \leq T$, we have that $\Lambda_i = \hat{F}_0(\Lambda_{i-1})$. By Lemma 13, we have that

$$(1 - 3Hx)F_0(\Lambda_{i-1}) \preceq \Lambda_i \preceq (1 + 3Hx)F_0(\Lambda_{i-1}) + 4Hx\Lambda_{i-1}. \tag{33}$$

Then we prove by induction that

$$\Lambda_i \preceq C_i\Lambda_{i-1}, \tag{34}$$

where $C_i = (1 + 11Hx)^i$ for $1 \leq i \leq T$. For $i = 1$, we learn that $\Lambda_0 = \mathbf{I}$ and $\Lambda_1 = \hat{F}_0(\mathbf{I}) \preceq (1 + 3Hx)F_0(\mathbf{I}) + 4Hx\mathbf{I} \preceq (1 + 7Hx)\mathbf{I}$. For $i \geq 2$, by the induction and the fact that $F_0(a\Lambda) \leq aF_0(\Lambda)$ for $a \geq 1$, we have that

$$F_0(\Lambda_{i-1}) \preceq F_0\left(C_{i-1}\Lambda_{i-2}\right) \preceq C_{i-1}F_0(\Lambda_{i-2}). \tag{35}$$

By (33) and (35), we have that

$$
\begin{aligned}
\Lambda_i &\preceq (1 + 3Hx)F_0(\Lambda_{i-1}) + 4Hx\Lambda_{i-1} \\
&\preceq (1 + 3Hx)C_{i-1}F_0(\Lambda_{i-2}) + 4Hx\Lambda_{i-1} \\
&\preceq \frac{(1 + 3Hx)C_{i-1}}{1 - 3Hx}\Lambda_{i-1} + 4Hx\Lambda_{i-1} \\
&\preceq ((1 + 7Hx)C_{i-1} + 4Hx)\Lambda_{i-1} \\
&\preceq C_i\Lambda_{i-1}.
\end{aligned}
$$

The proof of (34) is finished.

By the update rule, we learn that

$$
\Lambda_i \preceq (1 + 11Hx)^i\Lambda_{i-1} \preceq (1 + 11Hx)^i\Lambda_{i-1};
$$
$$
\Lambda_i + \frac{\zeta}{2x}\mathbf{I} \not\succeq \frac{1}{2}\Lambda_{i-1},
$$

Let $\check{\Lambda}_i = \Lambda_i + \frac{\zeta}{2x}\mathbf{I}$ for $i \geq 0$. Then we learn that

$$
\check{\Lambda}_i \preceq (1 + 11Hx)^i\check{\Lambda}_{i-1}, \qquad \check{\Lambda}_i \not\succeq \frac{1}{2}\check{\Lambda}_{i-1}, \qquad \check{\Lambda}_i \succeq \frac{\zeta}{2x}\mathbf{I}.
$$

As a result, the maximal eigenvalue of $\check{\Lambda}_{i-1}^{-1/2}\check{\Lambda}_i\check{\Lambda}_{i-1}^{-1/2}$ is at most $(1 + 11Hx)^i$, while the minimal eigenvalue of $\check{\Lambda}_{i-1}^{-1/2}\check{\Lambda}_i\check{\Lambda}_{i-1}^{-1/2}$ is at most $\frac{1}{2}$. Then we have that

$$
\log(\det(\check{\Lambda}_i)) - \log(\det(\check{\Lambda}_{i-1})) \leq di\log(1 + 11Hx) - \log(2).
$$

By noting that $d\log(\zeta/2x) \leq \log(\det(\check{\Lambda}_i))$ and $\log(\det(\check{\Lambda}_0)) \leq d\log(1 + \zeta/2x)$, we learn that for any $1 \leq j \leq T$

$$
-d\log(2x/\zeta + 1) \leq \sum_{i=1}^{j} di\log(1 + 11Hx) - j\log(2) \leq 0.
$$

As a result, it holds that

$$
d\log(2x/\zeta + 1) \geq j\log(2) - \frac{j(j+1)}{2}d\log(1 + 11Hx)
$$

for any $1 \leq j \leq T$. Solving the quadratic inequality, we learn that $T \leq 10d\log\left(\frac{2x}{\zeta} + 1\right)$.

**Truncation probability.** By definition, we have $\Lambda_{\mathrm{end}} = \Lambda_T$. Note that $\Lambda_{\mathrm{end}} \succeq (1-3Hx)F_0(\Lambda_{\mathrm{end}})$ and $F_0(\Lambda_{\mathrm{end}}) = \mathbb{E}_\pi\left[\mathrm{T}(\phi_h\phi_h^\top, f_1\Lambda_{\mathrm{end}})\right]$. We then have that

$$
\mathbb{E}_\pi\left[\mathrm{Trace}\left(\mathrm{T}(\phi_h\phi_h^\top, f_1\Lambda_{\mathrm{end}})(\Lambda_{\mathrm{end}})^{-1}\right)\right] \leq \frac{d}{(1 - 3Hx)}.
$$

On the other hand, by noting that

$$
\mathrm{Pr}_\pi\left[\phi_h^\top(\Lambda_{\mathrm{end}})^{-1}\phi_h > f_1\right] \cdot f_1 \leq \mathbb{E}_\pi\left[\mathrm{Trace}\left(\mathrm{T}(\phi_h\phi_h^\top, f_1\Lambda_{\mathrm{end}})(\Lambda_{\mathrm{end}})^{-1}\right)\right] \leq \frac{d}{(1 - 3Hx)},
$$

we have

$$
\mathrm{Pr}_\pi\left[\phi_h^\top(\Lambda_{\mathrm{end}})^{-1}\phi_h > f_1\right] \leq \frac{d}{f_1(1 - 3Hx)}.
$$

$\square$

## C.8 Statement and Proof of Lemma 19

**Lemma 19.** *Recall that $z = \frac{100000\epsilon^2}{d^2 H^5}$. Let $\mathcal{D}_h = \{\phi_{h,i}, \tilde{s}_{h,j}, \lambda_{h,i}\}_{i=1}^N$ be the one sub-dataset in in Line 9, Algorithm 7. With probability $1 - \frac{\delta}{16m^2 H^2}$, it holds that*

$$\sum_{i=1}^N \lambda_{h,i}^2 \phi_{h,i} \phi_{h,i}^\top + z\mathbf{I} \succeq \frac{N}{8m} \cdot \check{\Lambda}_h.$$

*Proof.* Let $X_h^i$ and $Y_h^i$ be respectively the final value of $\Lambda$ and $\hat{F}_0$ in the $i$-th call of Algorithm 3 in Algorithm 2 for the $h$-th round. Let $\mathbf{I}_h = \mathbb{I}\left[\phi_\tau(s_\tau, \pi_\tau(s_\tau))^\top \check{\Lambda}_\tau^{-1} \phi_\tau(s_\tau, \pi_\tau(s_\tau)) < 1, \forall 1 \le \tau \le h-1\right]$. By Lemma 13 it holds that

$$(1 + 3Hx)\mathbb{E}_{\pi^{i,h}}\left[\mathbf{I}_h \mathbf{T}(\phi_h \phi_h^\top, f_1 X_h^i)\right] + 4Hx X_h^i + \frac{\zeta}{2x}\mathbf{I} \succeq Y_h^i + \frac{\zeta}{2x}\mathbf{I} \succeq \frac{1}{2}X_h^i$$

and

$$(1 + 3Hx)\mathbb{E}_{\pi^{i,h}}\left[\mathbf{I}_h \mathbf{T}(\phi_h \phi_h^\top, f_1 X_h^i)\right] + 6Hx Y_h^i + \frac{\zeta}{2x}\mathbf{I} \succeq Y_h^i + \frac{\zeta}{2x}\mathbf{I}.$$

Because $\check{\Lambda}_h \succeq \frac{1}{2}X_h^i$

$$\begin{aligned}
\mathbb{E}\left[\sum_{i=1}^N \lambda_{h,i}^2 \phi_{h,i}\phi_{h,i}^\top\right] &\succeq \frac{N}{2m}\sum_{j=1}^m \mathbb{E}_{\pi^{j,h}}\left[\mathbf{I}_h \mathbf{T}(\phi_h\phi_h^\top, f_1 X_h^j)\right] \\
&\succeq \frac{N}{2m}\cdot\sum_{j=1}^m \frac{1}{1+3Hx}\cdot\left((1-6Hx)Y_h^j\right) \\
&\succeq \frac{N}{2m}\cdot\sum_{j=1}^m \frac{1}{2}\bar{\Lambda}_h^j \\
&\succeq \frac{N}{2m}\cdot\left(\frac{1}{2}\check{\Lambda}_h - \zeta\mathbf{I}\right).
\end{aligned} \tag{36}$$

Also noting that $\lambda_{h,i}\phi_{h,i}\phi_{h,i}^\top \preceq f_1\check{\Lambda}_h$, using Lemma 9, we have that, with probability $1 - \frac{\delta}{16mH^2}$,

$$\begin{aligned}
\sum_{i=1}^N \lambda_{h,i}^2 \phi_{h,i}\phi_{h,i}^\top &\succeq \frac{1}{2}\mathbb{E}\left[\sum_{i=1}^N \lambda_{h,i}^2 \phi_{h,i}\phi_{h,i}^\top\right] - f_1\check{\Lambda}_h \log(16mH^2/\delta) \\
&\succeq \frac{N}{8m}\check{\Lambda}_h - \frac{N}{8xm}\zeta\mathbf{I} \\
&\succeq \frac{N}{8m}\check{\Lambda}_h - z\mathbf{I}.
\end{aligned} \tag{37}$$

The proof is completed by re-arranging (37). $\qquad\square$

## C.9 Proof of Lemma 5

Let $\Theta$ be a $dH$-dimensional grid with distance $\frac{\epsilon}{8dH}$. Let $\text{Proj}_\Theta(\cdot)$ be the projection function to $\Theta$ by projecting each dimension to the grid. It is obvious that if $\theta = \{\theta_h\}_{h\in[H]}$ satisfies that $\|\theta_h\|_2 \le d, \forall h \in [H]$, then $\|\text{Proj}_{\Theta,h}(\theta)\|_2 \le 2d, \forall h \in [H]$.

It suffices to show that for any kernel $\{\theta_h\}_{h\in[H]} \in \Theta$, the output policy is $\frac{3}{4}\epsilon$-optimal. Assume the conditions in Lemma 6 holds. Let $\check{M}$ be the final truncated MDP $M_H$. Then we have that

$$\max_\pi \text{Pr}_\pi\left[\exists h \in [H], \phi_h^\top \check{\Lambda}_h \phi_h > 1\right] \le H \cdot \frac{\epsilon}{8H^2} \le \frac{\epsilon}{8H}.$$

As a result, for any $\pi$ and reward function $r$ such that $\|r\|_\infty \le 1$, we have that

$$\left|\mathbb{E}_\pi\left[\sum_{h=1}^H r_h\right] - \mathbb{E}_{\pi,\check{M}}\left[\sum_{h=1}^H r_h\right]\right| \le \frac{\epsilon}{8}.$$

Fix reward kernel $\theta = \{\theta_h\}_{h \in [H]} \in \Theta$. We continue the analysis by assuming the ground MDP is $\check{M}$. Let $\pi$ be the returned policy and $\pi^*$ be the optimal policy. Let $\{V^*_{h,\theta}(s), Q^*_{h,\theta}(s,a)\}$ and $\{V^\pi_{h,\theta}(s), Q^\pi_{h,\theta}(s,a)\}$ be respectively the optimal value function and the value function of $\pi$. In particular, we let $V^*_{0,\theta} = V^*_{1,\theta}(s_{\text{ini}})$. Let $\{V_{h,\theta}(s), Q_{h,\theta}(s,a)\}$ be the value of $\{V_h(s), Q_h(s,a)\}$ in Algorithm 5 with input reward kernel as $\theta$. Let $V_{0,\theta} = V_{1,\theta}(s_{\text{ini}})$ and $V^\pi_{0,\theta} = V^\pi_{1,\theta}(s_{\text{ini}})$. When $\theta$ is clear from the context, we omit $\theta$ in the subscript.

We then have that

$$V^*_0 - V^\pi_0 = (V^*_0 - V_0) + (V_0 - V^\pi_0). \tag{38}$$

We then prove by induction that $V^*_h(s) - V_h(s) \leq (H - h) \cdot \frac{\epsilon}{8H}$ for all $s \in \mathcal{S}$ and $h \in [H]$. The inequality is trivial for $h = H$. Now we assume it is correct for all $h \geq \ell$. Let $X_\tau = \sum_{i=1}^N \lambda^2_{\tau,i} \phi_{\tau,i} \phi^\top_{\tau,i} + z\mathbf{I}$ for $\tau \in [H]$. Recall that $\Phi(\xi)$ is an $\xi$-net of the $d$-dimensional unit ball. Fix $\phi \in \Phi(\xi)$ with $\|\phi\|_2 \leq 1$ and $V \in \mathbb{R}^{\mathcal{S}}$ with $\|V\|_\infty \leq H$. By Bernstein's inequality (1-dimensional case of Lemma 9), with probability $1 - \frac{\delta}{4H|\Phi(\xi)| \cdot |\Theta|}$, it holds that

$$\left| \phi^\top X_h^{-1} \sum_{i=1}^N \lambda^2_{h,i} \phi_{h,i} V(\tilde{s}_{h,i}) - \phi^\top \mu^\top_\tau V \right|$$

$$\leq 4\sqrt{\phi^\top X_\tau^{-1} \phi \log\left(\frac{4H|\Phi(\xi)| \cdot |\Theta|}{\delta}\right)} + 2 \max_i \sqrt{\phi^\top X_h^{-1} \phi \cdot \lambda^2_{h,i} \phi^\top_{h,i} X_h^{-1} \phi_{h,i} \log\left(\frac{4H|\Phi(\xi)| \cdot |\Theta|}{\delta}\right)}$$

$$\leq \sqrt{\frac{128m}{N} \log\left(\frac{4H|\Phi(\xi)| \cdot |\Theta|}{\delta}\right)} + \sqrt{\frac{32m}{N} \cdot \phi^\top X_h^{-1} \phi \log\left(\frac{4H|\Phi(\xi)| \cdot |\Theta|}{\delta}\right)}.$$

With a union bound over $\phi \in \Phi(\xi)$, we learn that, with probability $1 - \frac{\delta}{4H|\Theta|}$,

$$\left| \phi^\top X_h^{-1} \sum_{i=1}^N \lambda^2_{h,i} \phi_{h,i} V(\tilde{s}_{h,i}) - \phi^\top \mu^\top_h V \right| \leq 32\sqrt{\frac{mdH \log\left(\frac{dH}{\epsilon\delta}\right)}{N}} + \sqrt{\frac{128m}{N} \cdot \phi^\top X_h^{-1} \phi \cdot dH \log\left(\frac{dH}{\epsilon\delta}\right)}$$

$$\leq 32\sqrt{\frac{mdH \log\left(\frac{dH}{\epsilon\delta}\right)}{N}} + \frac{32mdH \log\left(\frac{dH}{\epsilon\delta}\right)}{N}$$

$$\leq \frac{\epsilon}{16H}$$

for any $\phi$ such that $\|\phi\|_2 \leq 1$ and $\phi^\top \check{\Lambda}_h \phi \leq 1$. Note that $V_{h+1,\theta}(\cdot)$ is determined by $\theta = \{\theta_h\}_{h \in [H]}$ and the sub-datasets after the $h$-th layer (non-inclusive). With a union bound over $\theta \in \Theta$, we learn that: with probability $1 - \frac{\delta}{4}$,

$$\left| \phi^\top X_h^{-1} \sum_{i=1}^N \lambda^2_{h,i} \phi_{h,i} V_{h+1,\theta}(\tilde{s}_{h,i}) - \phi^\top \mu^\top_h V_{h+1,\theta} \right| \leq \frac{\epsilon}{16H}$$

for any $\phi$ such that $\|\phi\|_2 \leq 1, \phi^\top \check{\Lambda}_h \phi \leq 1$ and $\theta \in \Theta$. Then we have that

$V^*_{\ell-1}(s) - V_{\ell-1}(s)$
$= Q^*_{\ell-1}(s, \pi^*_{\ell-1}(s)) - V_{\ell-1}(s)$
$\leq Q^*_{\ell-1}(s, \pi^*_{\ell-1}(s)) - Q_{\ell-1}(s, \pi^*_{\ell-1}(s))$

$$\leq P^\top_{\ell-1,s,\pi^*_{\ell-1}(s)}(V^*_\ell - V_\ell) + P^\top_{\ell-1,s,\pi^*_{\ell-1}(s)} V_\ell - \phi^\top_{\ell-1,s,\pi^*_{\ell-1}} X_{\ell-1}^{-1} \sum_{i=1}^N \lambda^2_{\ell-1} \phi_{\ell-1,i} V_\ell(\tilde{s}_{\ell,i}) + \frac{\epsilon}{16H}$$

$$\leq P^\top_{\ell-1,s,\pi^*_{\ell-1}(s)}(V^*_\ell - V_\ell) + \frac{\epsilon}{8H}$$

$$\leq \frac{\epsilon(H - h)}{8H}.$$

As a result, we learn that $V_0^* - V_0 \leq \frac{\epsilon}{8}$. For the second term $(V_0 - V_0^\pi)$ in (38), using similar arguments, we have that

$$V_0 - V_0^\pi = \mathbb{E}_\pi \left[ \sum_{h=1}^{H} Q_h(s_h, a_h) - \phi_h^\top \theta_h - P_{h,s_h,a_h}^\top V_{h+1}(s_h) \right]$$

$$\leq H \cdot \frac{\epsilon}{8H}$$

$$\leq \frac{\epsilon}{8}.$$

Putting all together, with probability $1 - \frac{\delta}{2}$, we have that $V_{0,\theta}^* - V_{0,\theta}^\pi \leq \frac{\epsilon}{4} \leq \frac{5\epsilon}{8}$ for all $\theta \in \Theta$. As a result, $\pi$ is at least a $\frac{3}{4}\epsilon$-optimal policy under the original MDP $M$. The proof is completed.

## D  Missing Algorithms

In this section, we present and explain the missing algorithms. Let $\mathrm{Range}_{[a,b]}(x) = a\mathbb{I}[x < a] + x\mathbb{I}[a \leq x \leq b] + b\mathbb{I}[x > b]$ for fixed $a, b \in \mathbb{R}$ and $x \in \mathbb{R}$.

**`Planning` (Algorithm 5).**  This algorithm is used to compute the optimal policy given a group of datasets. The planning method combines backward planning with linear regression. A key distinction is that the feature is clipped based on block matrices. Here $\Theta$ denotes a $dH$-dimensional grid with distance $\frac{\epsilon}{8dH}$, and $\mathrm{Proj}_\Theta(\cdot)$ denotes the projection operator to $\Theta$ by projecting each dimension to the grid. We refer the readers to Appendix C.9 for the effectiveness of this algorithm.

**`Planning-R` (Algorithm 6).**  This algorithm is used to compute the near-optimal policy given a fixed reward function. This algorithm is similar to `Planning` (Algorithm 5), except that the reward function is given as input (it is possible that the reward function is non-linear).

**`Policy-Execution` (Algorithm 7).**  This algorithm is used to collect multiple copies of the datasets. The efficiency of the collected dataset is explained in Lemma 19.

---

**Algorithm 5** `Planning`

---

**Input:** reward kernel $\theta = \{\theta_h\}_{h \in [H]}$, sub-datasets $\{\phi_{h,i}, \tilde{s}_{h,i}, \lambda_{h,i}\}_{i \in [N], h \in [H]}$, block matrices $\{\check{\Lambda}_h\}_{h \in [H]}$;

**Initialization:** $\theta \leftarrow \mathrm{Proj}_\Theta(\theta)$; $V_{H+1}(s) \leftarrow 0$ for all $s \in \mathcal{S}$;

**for** $h = H, H-1, \ldots, 1$ **do**

  **for** $(s, a) \in \mathcal{S} \times \mathcal{A}$; **do**

    $\phi \leftarrow \phi_h(s, a)$

$$Q_h(s,a) \leftarrow \begin{cases} \phi^\top \theta_h + \phi^\top \left( \sum_{i=1}^{N} \lambda_{h,i}^2 \phi_{h,i} \phi_{h,i}^\top + z\mathbf{I} \right)^{-1} \sum_{i=1}^{N} \lambda_{h,i}^2 \phi_{h,i} V_{h+1}(\tilde{s}_{h,i}), & \phi^\top \check{\Lambda}_h^{-1} \phi \leq 1; \\ 0, & \text{else}; \end{cases}$$

    $Q_h(s,a) \leftarrow \mathrm{Range}_{[0,H]}(Q_h(s,a));$

  **end for**

  **for** $s \in \mathcal{S}$ **do**

    $V_h(s) \leftarrow \max_a Q_h(s,a);$

    $\pi_h(s) \leftarrow \arg\max_a Q_h(s,a);$

  **end for**

**end for**

**return:** $\pi \leftarrow \{\pi_h\}_{h \in [H]}.$

---

## E  Computational Efficiency

In this section, we present the time complexity of our algorithms. In the rest of the analysis, we use the fact that the time cost of computing the inverse of a $d$-dimensional PSD matrix is $O(d^3)$.

---

**Algorithm 6** `Planning-R`

---

**Input:** horizon $h$, reward function $r$, sub-datasets $\{\phi_{\tau,i}, \tilde{s}_{\tau,i}, \lambda_{\tau,i}\}_{i \in [N], \tau \in [h-1]}$, block matrices $\{\check{\Lambda}_\tau\}_{\tau \in [h-1]}$;

$V_h(s) \leftarrow \max_a r_h(s,a), \forall s \in \{\tilde{s}_{h-1,i}\}_{i \geq 1}$;

**for** $\tau = h-1, h-2, \ldots, 1$ **do**

$\quad X_\tau \leftarrow \sum_{i=1}^N \lambda_{\tau,i}^2 \phi_{\tau,i} \phi_{\tau,i}^\top + z\mathbf{I}$;

$\quad$ **for** $s \in \{\tilde{s}_{\tau-1,i}\}_{i \geq 1}, a \in \mathcal{A}$ **do**

$\qquad \phi \leftarrow \phi_\tau(s,a)$;

$$Q_\tau(s,a) \leftarrow \begin{cases} \phi^\top X_\tau^{-1} \sum_{i \geq 1} \phi_{\tau,i} V_{\tau+1}(\tilde{s}_{\tau+1,i}) + 32\sqrt{\frac{md\log\left(\frac{dH}{\epsilon\delta}\right)}{N}} + \frac{32md\sqrt{f_1}\log\left(\frac{dH}{\epsilon\delta}\right)}{N}, & \phi^\top \check{\Lambda}_\tau^{-1}\phi \leq 1; \\ 0, & \text{else} \end{cases}$$

$\qquad Q_\tau(s,a) \leftarrow \text{Range}_{[0,1]}(Q_\tau(s,a))$;

$\quad$ **end for**

$\quad$ **for** $s \in \{\tilde{s}_{\tau-1,i}\}_{i \geq 1}$ **do**

$\qquad V_\tau(s) = \max_a Q_\tau(s,a)$;

$\qquad \pi_\tau(s) = \arg\max_a Q_\tau(s,a)$;

$\quad$ **end for**

**end for**

$V_0 \leftarrow V_1(s_{\text{ini}})$;

**return:** $\{V_0, \pi\}$

---

---

**Algorithm 7** `Policy-Execution`

---

1: **Input** $h$, $\{\pi^{i,h}\}_{i=1}^m$, $\check{\Lambda}_h$ :

2: $\pi \leftarrow \text{uniform}(\{\pi^{i,h}\}_{i=1}^m)$;

3: **for** $\tau = 1, 2, \ldots, H$ **do**

4: $\quad$ **for** $z = 1, 2, \ldots, 2m^2 + 1$ **do**

5: $\qquad$ **for** $j = 1, 2, \ldots, N$ **do**

6: $\qquad\quad$ Run $\pi$ to observe the feature $\phi_{h,j}$ and the next state $\tilde{s}_{h,j}$;

7: $\qquad\quad \lambda_{h,j} \leftarrow \min\left\{\sqrt{\frac{f_1}{\phi_{h,j}^\top \check{\Lambda}_h^{-1} \phi_{h,j}}}, 1\right\}$;

8: $\qquad$ **end for**

9: $\qquad \mathcal{D}_h^\tau(z) \leftarrow \{\phi_{h,j}, \tilde{s}_{h,j}, \lambda_{h,j}\}_{j=1}^N$;

10: $\quad$ **end for**

11: $\quad \mathcal{D}_h^\tau \leftarrow \{\mathcal{D}_h^\tau(z)\}_{z=1}^{2m^2+1}$

12: **end for**

13: **return** : $\mathcal{D}_h \leftarrow \{\mathcal{D}_h^\tau\}_{\tau=1}^H$.

---

**Truncated-Matrix-Eval (Algorithm 4).** Firstly, the truncation operator $\mathtt{T}(\cdot)$ could be implemented with time $O(d^3)$. Then the total computational cost of this algorithm is bounded by $O(H(Nd^2 + d^3)) = O(NHd^2)$.

**Matrix-Eval (Algorithm 3).** The computational cost of this algorithm is at most $O(m)$ multiplies that of $\mathtt{Truncated\text{-}Matrix\text{-}Eval}$ (Algorithm 4), which is $O(mNHd^2)$.

**Planning (Algorithm 5) and Planning-R (Algorithm 6).** These two algorithms shares similar structure, with computational cost $O(HANd^2)$ to compute the action give the current state.

**Policy-Design (Algorithm 2).** The computational cost of this algorithm is at most $O(m)$ multiplies that of $\mathtt{Matrix\text{-}Eval}$ (Algorithm 3) and $\mathtt{Planning\text{-}R}$ (Algorithm 6), which is bounded by $O(m^2 NHd^2)$.

**Policy-Execution (Algorithm 7).** The time cost of this algorithm is simply $O(m^4 N^2 H^2 A d^2)$.

**Exploration (Algorithm 1).** By the above results, the total computation cost of this algorithm is $O(m^4 N^2 H^2 d^2 A) = \tilde{O}\left(\frac{d^{32} H^{28} A}{\epsilon^{10}}\right)$.

