# OpenReview forum: "Deployment Efficient Reward-Free Exploration with Linear Function Approximation"
_NeurIPS.cc/2025/Conference — NeurIPS 2025 poster_

### Official Review · Reviewer_tYnN · 2025-06-19

**Clarity:** 2
**Significance:** 3
**Originality:** 3
**Rating:** 4
**Confidence:** 4

**Summary:**

In this paper, the authors study deployment-efficient reward-free exploration with linear function approximation, where the goal is to explore a linear Markov Decision Process (MDP) without revealing the reward function, while minimizing the number of distinct policies implemented during learning. For the task, the authors design a provably efficient algorithm with near optimal deployment efficiency while the sample complexity has polynomial dependence on relevant parameters. Compared to previous algorithms, this work removes a crucial assumption of reachability, directly addressing an open problem from prior work.

**Questions:**

Please refer to the weaknesses.

**Ethical Concerns:**

["NO or VERY MINOR ethics concerns only"]

**Limitations:**

Yes.

**Quality:**

3

**Strengths And Weaknesses:**

Strengths:
1. The problem of deployment efficient RL is important and interesting.
2. The technical results in this paper are solid, while the paper provides lots of novel techniques.
3. This paper removes the assumption of reachability, which is a strong assumption in previous works.

Weaknesses:
1. The sample complexity upper bound does not appear in the main part, and the bound is far from optimal. The bound is better than previous results only when the reachability coefficient in previous papers is very close to 0. While I understand the difficulty to remove the reachability assumption, it will be helpful if there are more discussions about the difficulty to improve the current sample complexity.

2. The proof overview in the main part is not very informative. The intuition of the three statements in Lemma 6 is not clear and requires more explanation. A proof sketch (even in the appendix) showing the amount of data for different parts of the algorithm (e.g. policy design, matrix evaluation) and how does these data combine to the whole sample complexity would be helpful.

3. I would suggest the authors to polish the writing and check for the typos in the main part. E.g., in line 7 of Algorithm 4 and (i) of Lemma 6, should the $\Lambda$ be $\Lambda^{-1}$?

Overall, I think removing the reachability assumption is an important contribution. Therefore, I slightly lean towards acceptance of the paper.

---

> ### Author Rebuttal · Authors · 2025-07-31
>
> We thank the reviewer for the insightful comments. Below we provide our responses to each point.
>
> **Regarding the bad dependencies on $d, H$ and $1/\epsilon.$**
>
> The main reason of the bad dependency is that: The main algorithm (Algorithm 1) involves two iterative processes, and since the results of different iterations all rely on the same offline dataset, these results are subtly coupled with each other. To break the statistical coupling, we simply make independent copies of the offline datasets by following the exploration policy and repeatedly sampling trajectories  with fresh randomness.
>
>  One potential approach to improving the sample complexity is to  prove concentration inequalities (e.g., Lemma 17) through union bound arguments. The major difficulty in this approach is that the complexity of the offline planning policy in Algorithm 3 and 4 could be high.
>
>
>
> **Regarding the proof sketch of the sample complexity.**
>
> Thanks for the suggestion. In Section 6, we present the framework for proving the main theorem. The most complicated components are the proofs of Lemmas 6 and 15, for which we will provide high-level proof sketches in the next revision.
>
> Here, we present a summary of the sample complexity for each component of the algorithm. We will provide the details in Appendix E.
>
> We begin by counting the number of sub-datasets required throughout the learning process. Then the sample complexity (total number of episodes) is obtained by multiplying this number $N = \frac{10^{9}d^7H^7\log\left(\frac{dH}{\epsilon\delta}\right)}{\epsilon^3}$.
>
> *Algorithm 4* (Truncated-Matrix-Eval): This algorithm requires at most $H$ sub-datasets.
>
> *Algorithm 3* (Matrix-Eval): This algorithm makes $O(m)$ calls to Algorithm 4, each requiring at most $O(Hm)$ sub-datasets.
>
> The effectiveness about Algorithm 3 (including Algorithm 4) is provided in Lemma 18, which means we could achieve efficient truncation with recursive truncated planning.
>
> *Algorithm 5* (Planning) and *Algorithm 6* (Planning-R): These two algorithms shares similar structure, which requires at most $H$ sub-datasets.
>
> *Algorithm 2* (Policy-Design): This algorithm makes $O(m)$ calls to Algorithm 3  and  Algorithm 5, which requires at most $O(m^2H)$ sub-datasets.
>
> *Algorithm 1* (Exploration): This algorithm makes $H$ calls to Algorithm 2, which requires at most $O(m^2H^2)$ sub-datasets.
>
> Putting all together, the total number of episodes is $O(m^2H^2N)=\tilde{O}\left( \frac{d^{15}H^{14}}{\epsilon^5}\right)$.
>
>
>
> **Regarding the typos.**
>
>  Thanks for pointing out the typos and we will fix accordingly in the next revision.

---

> > ### Comment · Reviewer_tYnN · 2025-08-04
> >
> > Thanks for the detailed reply. I will keep my score.

---

### Official Review · Reviewer_Ty3N · 2025-06-21

**Clarity:** 2
**Significance:** 2
**Originality:** 2
**Rating:** 4
**Confidence:** 4

**Summary:**

This paper studies reward free exploration problem in linear MDP problem. In particular, we assume a linear MDP setting without the knowledge of the reward function. The goal is to accumulate enough data to find a good estimate of the underlying MDP. This estimation then will be used in the next step to find an estimation of the optimal policy corresponding to an arbitrary reward function. The authors develop this algorithm, and show that their algorithm requires only a polynomial number of samples with respect to the problem parameters.

**Questions:**

- What is "z" in Algorithm 5?
- Can you provide a sample complexity (with exact dependency on parameters of the MDP) for tabular MDP setting?

**Ethical Concerns:**

["NO or VERY MINOR ethics concerns only"]

**Final Justification:**

The authors conducted experiment, and claim that their algorithm outperforms prior work. If paper gets accepted, I expect to see the result of the experiment in the final version of the paper.

**Limitations:**

yes

**Paper Formatting Concerns:**

-	The order of writing in the paper needs to be changed. It is not helpful to have all the discussion in pages 5 and 6 before even introducing the algorithm.
- When discussing about infrequent directions, it is better to first define it clearly, and explain the assumption that appears in the prior work. Just mentioning it in words is not sufficient, and would make the reader confused.
-	The paper needs a rewriting since there are several grammatical errors.

**Quality:**

3

**Strengths And Weaknesses:**

The main strength of the paper is to remove the infrequent direction assumption in the prior work. In particular, this work does not assume a uniform lower bound on the eigen values of $\Lambda_h$ matrix over all the horizons. Instead, they devise algorithm 3, which helps on achieving a good data set, with at most H number of deployments (rather than exponential number of deployments with respect to H, which might happen since the lower bound on the distribution over states can decrease exponentially with respect to the horizon).
The main weakness is that in terms of problem formulation is relatively similar to the prior work, and the work is summarized in just a theorem with polynomial dependency on \epsilon, H, d, and log(1/$\delta$). This polynomial dependency can be very bad, and no lower bounds or simulations are provided to support the result.

---

> ### Author Rebuttal · Authors · 2025-07-31
>
> We are grateful to the reviewer for the constructive feedback. Below we present our response.
>
> **Regarding the bad polynomial dependency.**
>
> Indeed our current sample complexity is far from being optimal. The main reason of the bad dependency is that: The main algorithm (Algorithm 1) involves two iterative processes, and since the results of different iterations all rely on the same offline dataset, these results are subtly coupled with each other. To break the statistical coupling, we simply make independent copies of the offline datasets by following the exploration policy and repeatedly sampling trajectories  with fresh randomness.
>
>  One potential approach to improving the sample complexity is to  prove concentration inequalities (e.g., Lemma 17) through union bound arguments. The major difficulty in this approach is that the complexity of the offline planning policy in Algorithm 3 and 4 could be high.
>
> **Regarding the lower bound.**
>
> While the precise trade-off between sample complexity and deployment complexity remains unknown, Theorem 3.2 in Huang et al., 2022 provides an $\Omega(H/\log_d(1/\epsilon))$ lower bound for the sample complexity under polynomial sample complexity constraints.
>
> **Regarding the numerical experiments.**
>
> The primary goal of this work is to eliminate dependence on reachability in both sample complexity and deployment complexity. To ensure our algorithm outperforms reachability-dependent approaches, we require the reachability parameter to be exponentially small. However, this leads to highly artificial MDP constructions. In practice, features are typically learned via neural networks, making it unclear how to test reachability parameters in such settings. Since different features yield different reachability values, evaluating the algorithm in a natural, practical environment becomes challenging.
>
> **Regarding $z$ in Algorithm 5.**
>
> We set $z = \frac{100000\epsilon^2}{d^4H^5}$. Please refer to line 781 for the parameter settings.
>
> **Regarding the tabular case.**
>
> Let $S$ and $A$ be respectively the sizes of the state space and the action space.
> By the method in (Zhang et al., 2022), one could reach the asymptotically optimal $\tilde{O}(\frac{SAH^3}{\epsilon^2})$ sample complexity (in episodes) with $H+1$ deployments.
> In this approach, we use the first $H$ deployments (which corresponds to $H$ batches in Zhang et al., 2022) to learn an approximate transition model $\tilde{P}$ using $\mathrm{poly}(S,A,H)$ episodes. Then we  design an exploration policy $\pi'$ with minimax coverage under the approximate model $\tilde{P}$ such that
> $$\max_{\pi} \sum_{s,a,h} \frac{d_{\pi}(h,s,a)} {d_{\pi'}(h,s,a)}  = O(SAH).$$
> Here $d_{\pi}(h,s,a) = \mathbb{E}_{\pi,\tilde{P}} \mathbb{I}(s_h=s,a_h=a)$ denotes the occupancy distribution of the policy $\pi$ under the transition $\tilde{P}$.
>   By running $\pi'$  for $O(\frac{SAH^3}{\epsilon^2})$ episodes, we could learn an $\epsilon$ optimal policy.
> Similar approaches have also been explored by Qiao et al. (2022).
>
> **References:**
>
> Huang et al., 2022: Towards deployment-efficient reinforcement learning: lower bound and optimality.
>
> Zhang et al., 2022: Near-Optimal Regret Bounds for Multi-batch Reinforcement Learning.
>
> Qiao et al., 2022: Sample-efficient reinforcement learning with $\log\log(t)$ switching cost.

---

> > ### Comment · Reviewer_Ty3N · 2025-08-06
> >
> > Thank you for the detailed response.
> > For experimental results, can you test your algorithm on a small simple MDP with a few state and actions?

---

> > > ### Author Response · Authors · 2025-08-08
> > >
> > > We would like to thank the reviewer for the constructive comments. As suggested by the reviewer, we conducted experiments on the following small and simple MDP.
> > >
> > > The horizon length is $H = 7$, meaning that there are $7$ levels in the MDP. The action space contains three actions: $a_1, a_2, a_3$. There is a single initial state $s_0$ at the first level. In the middle levels, i.e., $2 \le h \le 6$, there are three different states: $s_1, s_2, s_3$. At the last level $h = 7$, there are two terminal states: $s_{\text{win}}$ and $s_{\text{lose}}$.
> > >
> > > The transitions are defined as follows:
> > > * Choosing action $a_i$ at the initial state $s_0$ transitions to state $s_i$ at $h = 2$.
> > > * For $2 \le h \le 5$,  the transitions are independent of the action chosen:
> > >    *  $s_1$ and $s_2$ transition back to themselves at the next level.
> > >    *  $s_3$ transitions to $s_2$ and $s_3$ with equal probability, i.e., $1/2$.
> > > * At level $h = 6$, the transitions are probabilistic and independent of the action chosen:
> > >     * State $s_1$ transitions to $s_{\text{win}}$ with probability $1/2 + \epsilon$ and $s_{\text{lose}}$ with probability $1/2 - \epsilon$.
> > >     * States $s_2$ and $s_3$ transition to $s_{\text{win}}$ with probability $1/2 - \epsilon$ and $s_{\text{lose}}$ with probability $1/2 + \epsilon$.
> > >
> > > We set $\epsilon = 0.05$ in our experiments. The agent receives a reward of $1000$ at $s_{\text{win}}$, and a reward of $-1000$ at $s_{\text{lose}}$. The reward values for all other states are set to $0$.
> > >
> > > We compare the new algorithm in our paper with Algorithm 2 from the prior work by Huang et al. [2022]. This is a fair comparison, as these two algorithms have the same deployment complexity. In our experiments, we vary the number of trajectories per deployment (denoted as $N$) and report the expected total reward of the returned policy, averaged over 1000 repetitions. Note that in the underlying MDP, the smallest state-reaching probability is $2^{-4} =0.0625$, which is not exceptionally small. However, even in this setting, we observe the following results.
> > >
> > > | $N$     | Our algorithm | Huang et al. |
> > > |---------|---------------|--------------|
> > > | 30      | 1.86          | -4.14        |
> > > | 300     | 1.92          | -3.66        |
> > > | 3000    | 4.2           | -0.84        |
> > > | 30000   | 9.36          | 4.52         |
> > > | 300000   | 10.00          | 9.54         |
> > >
> > >
> > > From the results, it is clear that our new algorithm outperforms the algorithm by Huang et al. [2022] across a wide range of $N$, as long as $N$ is not very large. This aligns with our theoretical analysis, as our algorithm quickly identifies $s_3$ as an infrequent state (dimension), and will not waste samples trying to reach $s_3$. On the other hand, such a mechanism does not exist in the previous algorithm by Huang et al. [2022] (as it assumes that all directions (states) are reachable). Consequently, it will waste a significant amount of samples on infrequent states (dimensions).
> > >
> > > We note that it is possible to further increase the performance gap between our algorithm and that of Huang et al. [2022], by possibly adding more infrequent states (dimensions) and more levels. Given the time constraints of the author-reviewer discussion period, we will add more discussion on these experiments to our final version.

---

> > > > ### Author Response · Authors · 2025-08-09
> > > >
> > > > Dear Reviewer Ty3N,
> > > >
> > > > We sincerely thank you again for your valuable feedback, which has greatly helped us improve the quality of our work. As the discussion deadline is approaching, we kindly request you to review our rebuttal and our response on experiments. We also sincerely ask you to consider increasing the score if our responses has resolved the main concerns.
> > > >
> > > > Thank you again for your time and effort!

---

### Official Review · Reviewer_ktTz · 2025-07-01

**Clarity:** 4
**Significance:** 3
**Originality:** 3
**Rating:** 5
**Confidence:** 2

**Summary:**

The paper proposes an algorithm for reward-free exploration in linear MDPs. Its main result is a nearly optimal deployment complexity (i.e., the number of different exploration policies used to collect data) of $O(H)$, and a polynomial (though suboptimal) sample complexity. Its main technical novelty is providing such guarantees without assumptions on the reachability coefficients of the MDP.

**Questions:**

- From line 782, the sample complexity of your algorithm is $O(d^{15} H^{15} / \epsilon^5)$, right? This information would be useful in the main text. An interesting direction for future work could be improving this sample complexity. Hence, it would be helpful if you could summarise the main causes of sub-optimality there!

- In algorithm 1, there seems to be exactly $H$ policy deployments. Why is the deployment complexity in Theorem 4 $O(H)$ and not exactly $H$?

- Out of curiosity, is there any result showing whether sample complexity and deployment complexity are conflicting objectives (i.e., whether both could be near-optimal)?

- I understand that this is a theoretical RL paper, but even in that case, I believe that experiments (on toy environments at least) are important. For example, one important contribution of the paper is obtaining guarantees without reachability assumptions. From what I understood, this is also a consequence of algorithmic improvements (not only proof techniques), so it would be interesting to see if (i) your algorithm works in a toy MDP with hard-to-reach states; (ii) if previous algorithms that require reachability assumptions get stuck in that case. I also understand that, in order for it to be feasible to run experiments, some theoretical constants (often too large) need to be relaxed empirically, but that shouldn’t be difficult, and would illustrate the main algorithmic ideas that make exploration work.

**Ethical Concerns:**

["NO or VERY MINOR ethics concerns only"]

**Final Justification:**

I think the paper brings a relevant theoretical contribution to the field. I increase my rating from 4 to 5 following the authors responses, and experimental validation on a toy experiment.

**Limitations:**

Yes.

**Paper Formatting Concerns:**

None.

**Quality:**

3

**Strengths And Weaknesses:**

## Strengths

- The paper is organised and written clearly.

- The authors provide the first nearly optimal algorithm for their setting without making assumptions on reachability coefficients of the MDP.

## Weaknesses

- Lack of experiments (even on toy MDPs).

- According to Assumption 2, the considered linear MDP setting assumes a finite number of states, unlike [Jin et al. 2019](https://arxiv.org/pdf/1907.05388).

## Suggestions

- A table including the deployment complexity and the sample complexity of your algorithm and related work would be useful (even if in the appendix).

---

> ### Author Rebuttal · Authors · 2025-07-31
>
> We appreciate the reviewer's valuable comments. We respond to each comment below.
>
> **Regarding the numerical experiments.**
>
>   We thank the reviewer for the suggestion. The primary goal of this work is to eliminate dependence on reachability in both sample complexity and deployment complexity. To ensure our algorithm outperforms reachability-dependent approaches, we require the reachability parameter to be exponentially small. However, this leads to highly artificial MDP constructions. In practice, features are typically learned via neural networks, making it unclear how to test reachability parameters in such settings. Since different features yield different reachability values, evaluating the algorithm in a natural, practical environment becomes challenging.
>
> **Regarding the hardness in improving the sample complexity.**
>
> You are right that the sample complexity of our algorithm is $O(d^{15}H^{15}/\epsilon^5)$.
>  In the next revision, we will provide the precise sample complexity in the theorem statement.
>
> The main reason of the bad dependency is that: The main algorithm (Algorithm 1) involves two iterative processes, and since the results of different iterations all rely on the same offline dataset, these results are subtly coupled with each other. To break the statistical coupling, we simply make independent copies of the offline datasets by following the exploration policy and repeatedly sampling trajectories  with fresh randomness.
>
>  One potential approach to improving the sample complexity is to  prove concentration inequalities (e.g., Lemma 17) through union bound arguments. The major difficulty in this approach is that the complexity of the offline planning policy in Algorithm 3 and 4 could be high.
>
>
> **Regarding the trade-off between deployment efficiency and sample efficiency.**
>
> By Theorem 3.2 in Huang et al., 2022, achieving polynomial sample complexity requires at least $cH/\log_d(1/\epsilon)$ deployments for some constant c > 0. Any deployment budget below this threshold results in super-polynomial sample complexity. With $O(H)$ deployments, our algorithm achieves polynomial sample complexity. Moreover, with $\tilde{O}(dH)$ deployments, one could learn an $\epsilon$ optimal policy with a near optimal sample complexity of $\tilde{O}(\frac{d^2H^3}{\epsilon^2})$ with the help of online learning methods (He et al., 2022).
>
> The remaining challenge is to determine the deployment complexity threshold for achieving near-optimal sample complexity. Specifically, we ask:
> ***Are $\Omega(dH)$ deployments necessary to attain the near-optimal sample complexity of $\tilde{O}(\frac{d^2H^3}{\epsilon^2})$?***
> We leave this problem as an open question for future research.
>
> **Regarding the finite state space.**
>
> We are sorry for the confusion.  We will redefine $\mu$ as $d$ measures $(\mu_1,\mu_2,\ldots, \mu_d)$ over the state space $\mathcal{S}$ as in Jin et al., 2019. We emphasize that our algorithm and theoretical analysis hold even for infinite state spaces.
>
>
> **About the exact number of deployments.**
>
> The number of deployments is exactly $H$ assuming the initial state is fixed as $s_1$. Without this assumption, we need $H+1 $ deployments. We will clarify this in the next revision.
>
> **Regarding the table to summarize the related works.**
>
> We thank the reviewer for this suggestion and will include the table in the next revision.
>
>
>
> **References:**
>
> Huang et al., 2022: Towards deployment-efficient reinforcement learning: lower bound and optimality.
>
> He et al., 2022:  Nearly minimax optimal reinforcement learning for linear Markov Decision Processes.
>
> Jin et al., 2019: Provably efficient reinforcement learning with linear function approximation.

---

> > ### Comment · Reviewer_ktTz · 2025-08-06
> >
> > Thank you for the clarifications! It's good to know that the guarantees apply to infinite state spaces.
> > I have no other concerns and believe that the paper brings a relevant theoretical contribution to the field. I decided to keep my rating (4), as I believe that the paper would have a higher impact if it brought algorithmic ideas that could be used in practical implementations.

---

> > > ### Author Response · Authors · 2025-08-08
> > >
> > > As suggested by the reviewer, we conducted experiments on the following small and simple MDP to illustrate the main algorithmic ideas.
> > >
> > > The horizon length is $H = 7$, meaning that there are $7$ levels in the MDP. The action space contains three actions: $a_1, a_2, a_3$. There is a single initial state $s_0$ at the first level. In the middle levels, i.e., $2 \le h \le 6$, there are three different states: $s_1, s_2, s_3$. At the last level $h = 7$, there are two terminal states: $s_{\text{win}}$ and $s_{\text{lose}}$.
> > >
> > > The transitions are defined as follows:
> > > * Choosing action $a_i$ at the initial state $s_0$ transitions to state $s_i$ at $h = 2$.
> > > * For $2 \le h \le 5$,  the transitions are independent of the action chosen:
> > >    *  $s_1$ and $s_2$ transition back to themselves at the next level.
> > >    *  $s_3$ transitions to $s_2$ and $s_3$ with equal probability, i.e., $1/2$.
> > > * At level $h = 6$, the transitions are probabilistic and independent of the action chosen:
> > >     * State $s_1$ transitions to $s_{\text{win}}$ with probability $1/2 + \epsilon$ and $s_{\text{lose}}$ with probability $1/2 - \epsilon$.
> > >     * States $s_2$ and $s_3$ transition to $s_{\text{win}}$ with probability $1/2 - \epsilon$ and $s_{\text{lose}}$ with probability $1/2 + \epsilon$.
> > >
> > > We set $\epsilon = 0.05$ in our experiments. The agent receives a reward of $1000$ at $s_{\text{win}}$, and a reward of $-1000$ at $s_{\text{lose}}$. The reward values for all other states are set to $0$.
> > >
> > > We compare the new algorithm in our paper with Algorithm 2 from the prior work by Huang et al. [2022]. This is a fair comparison, as these two algorithms have the same deployment complexity. In our experiments, we vary the number of trajectories per deployment (denoted as $N$) and report the expected total reward of the returned policy, averaged over 1000 repetitions. Note that in the underlying MDP, the smallest state-reaching probability is $2^{-4} =0.0625$, which is not exceptionally small. However, even in this setting, we observe the following results.
> > >
> > > | $N$     | Our algorithm | Huang et al. |
> > > |---------|---------------|--------------|
> > > | 30      | 1.86          | -4.14        |
> > > | 300     | 1.92          | -3.66        |
> > > | 3000    | 4.2           | -0.84        |
> > > | 30000   | 9.36          | 4.52         |
> > > | 300000   | 10.00          | 9.54         |
> > >
> > >
> > > From the results, it is clear that our new algorithm outperforms the algorithm by Huang et al. [2022] across a wide range of $N$, as long as $N$ is not very large. This aligns with our theoretical analysis, as our algorithm quickly identifies $s_3$ as an infrequent state (dimension), and will not waste samples trying to reach $s_3$. On the other hand, such a mechanism does not exist in the previous algorithm by Huang et al. [2022] (as it assumes that all directions (states) are reachable). Consequently, it will waste a significant amount of samples on infrequent states (dimensions).
> > >
> > > We note that it is possible to further increase the performance gap between our algorithm and that of Huang et al. [2022], by possibly adding more infrequent states (dimensions) and more levels. Given the time constraints of the author-reviewer discussion period, we will add more discussion on these experiments to our final version.

---

> > > > ### Comment · Reviewer_ktTz · 2025-08-08
> > > >
> > > > Thank you, I believe that this experimental validation is a very nice sanity check for the novel theoretical ideas. Also, if you needed any relaxation on the theoretical constants to make it work, it would be nice to mention it in the appendix.
> > > >
> > > > Given this validation, I decided to increase my score to 5.

---

### Official Review · Reviewer_JxHB · 2025-07-03

**Clarity:** 3
**Significance:** 2
**Originality:** 2
**Rating:** 4
**Confidence:** 3

**Summary:**

This paper studies reward-free reinforcement learning under linear model, and aims to design deployment efficient algorithm, i.e. the number of exploration policy used is significantly less than the number of online episodes. The paper proposed a new algorithm that only needs $O(H)$ exploration policies to output an $\epsilon$-optimal policy at the planning phase, where H is the horizon length.

**Questions:**

1. Can authors discuss the tradeoff between deployment efficiency and the sample efficiency. Is there any fundamental hardness here (e.g. we cannot achieve deployment efficiency and sample efficiency at the same time) or is it because the technical tools we have so far have limitations. (While there is some discussion after thm4, it seems that it is about the reachability parameter rather than the deployment complexity and the sample complexity themselves if I understand correctly.)

**Ethical Concerns:**

["NO or VERY MINOR ethics concerns only"]

**Final Justification:**

The authors have addressed my major concerns. I decide to keep my score.

**Limitations:**

Yes

**Paper Formatting Concerns:**

No further concerns.

**Quality:**

3

**Strengths And Weaknesses:**

Strengths:
1. The theoretical result seems novel and it shows that the number of exploration policies is independent of the sub-optimal gap $\epsilon$.
2. The algorithm design is new in terms of the handling infrequent visits in the linear MDP setting.

Weaknesses:
1. While the algorithm design is indeed novel, and provide insights on how to design deployment efficient algorithms in linear or low-rank RL, the problem is limited to linear MDPs with known features, thus the significance is reduced.
2. It is better to provide a comparison table of the results of related works showing the deployment complexity, sample complexity and the MDP setting.

---

> ### Author Rebuttal · Authors · 2025-07-31
>
> We thank the reviewer for the valuable comments. Below we provide our responses to each point.
>
> **Regarding the linear MDP assumption.**
>
> While our algorithm is currently limited to linear MDPs, extending these results to low-rank MDPs or MDPs with linear Bellman completeness would be an interesting direction for future work.
>
>
>
>
>
>
>  **Regarding  the table to summarize the related works.**
>
>  Thanks for the suggestion. We will include this table in the next revision. The most important related works include: (i) Huang et al., 2022, which achieves $O(H)$ deployment complexity with the reachability  assumption; (ii) Zhao et al., 2023, which obtains $O(dH)$ deployment complexity for general function approximation.
>
> **Regarding  the trade-off between deployment efficiency and sample efficiency.**
>
> By Theorem 3.2 in Huang et al., 2022, achieving polynomial sample complexity requires at least $cH/\log_d(1/\epsilon)$ deployments for some constant c > 0. Any deployment budget below this threshold results in super-polynomial sample complexity. With $O(H)$ deployments, our algorithm achieves polynomial sample complexity. Moreover, with $\tilde{O}(dH)$ deployments, one could learn an $\epsilon$ optimal policy with a near optimal sample complexity of $\tilde{O}(\frac{d^2H^3}{\epsilon^2})$ with the help of online learning methods (He et al., 2022).
>
> The remaining challenge is to determine the deployment complexity threshold for achieving near-optimal sample complexity. Specifically, we ask:
>  ***Are $\Omega(dH)$ deployments necessary to attain the near-optimal sample complexity of $\tilde{O}(\frac{d^2H^3}{\epsilon^2})$?*** When the reachability assumption holds, it could be possible to improve the deployment complexity to $H$ while maintaining near optimal sample complexity with existing techniques. However, when the reachability assumption does not hold, the technical tools we have so far fail to address  this problem.  We leave this problem as an open question for future research.
>
>
>
>
>
> **References:**
>
> Huang et al., 2022: Towards deployment-efficient reinforcement learning: lower bound and optimality.
>
> Zhao et al., 2023:  A nearly optimal and low-switching algorithm for reinforcement learning with general function approximation.
>
> He et al., 2022:   Nearly minimax optimal reinforcement learning for linear Markov Decision Processes

---

> > ### Comment · Reviewer_JxHB · 2025-08-05
> >
> > Thank you for your response and for addressing my concerns. I have no further questions and decide to keep my score.

---

### Decision · Program_Chairs · 2025-09-17

**Decision:**

Accept (poster)

**Comment:**

All reviewers agree that the paper is a solid theoretical contribution to the problem of deployment-efficient exploration with function-approximation.  It is the first efficient algorithm that handles the whole family of linear MDPs.  Existing work, e.g., Huang et al, and Qiao et al. while providing better dependence on parameters like H and d, their sample complexity (at least in finite sample) depends on certain "reachability" conditions.

The results are primarily technical, and the paper is recommended to be accepted as a theoretical work without experiments.   The authors should work on addressing all reviewer comments (including improving clarity) for the camera ready version.